# Effects of iron supplementation on cognitive development in school-age children: Systematic review and meta-analysis

**Befikadu Tariku Gutema** [1,2] *, **Muluken Bekele Sorrie**[1], **Nega Degefa Megersa**[3], **Gesila Endashaw Yesera**[3], **Yordanos Gizachew Yeshitila**[3], **Nele S. Pauwels**[4], **Stefaan De Henauw**[2], **Souheila Abbeddou** [2]

**1** School of Public Health, Arba Minch University, Arba Minch, Ethiopia, **2** Public Health Nutrition Unit, Department of Public Health and Primary Care, Ghent University, Ghent, Belgium, **3** School of Nursing, Arba Minch University, Arba Minch, Ethiopia, **4** Knowledge Centre for Health Ghent, Ghent University Hospital, Ghent, Belgium

* befikadutariku.gutema@ugent.be, befikadutariku2@gmail.com

**Data Availability Statement:** All relevant data are within the paper and its Supporting information files.

## Abstract

### Background

Iron deficiency is negatively associated with children's cognitive development. Evidence showed that iron supplementation improves cognitive development. Nearly 50% of anemia is caused by iron deficiency. Anemia affects more school-age children, at an age where their brain development continues. The aim of this systematic review and meta-analysis is to review the evidence from published randomized controlled trials to evaluate the effects of iron supplementation on cognitive development and function among school-age children.

### Method

Five databases including MEDLINE, EMBASE, Scopus, Web of Science and CENTRAL were used to search for articles on April 20th, 2021. The search was reconducted on October 13th, 2022 to retrieve new records. Studies were eligible if they included school children 6–12 years of age, were randomized controlled trials, and if they tested iron supplementation and measured cognitive development.

### Result

Thirteen articles were included in the systematic review. Overall, iron supplementation significantly improved intelligence (standardized mean difference, 95% confidence interval) (SMD 0.46, 95%CI: 0.19, 0.73, P<0.001), attention and concentration (SMD 0.44, 95%CI: 0.07, 0.81, P = 0.02) and memory (SMD 0.44, 95%CI: 0.21, 0.67, P <0.001) of school-age children. There was no significant effect of iron supplementation on school achievement of school-age children (SMD 0.06, 95%CI: -0.15, 0.26, P = 0.56). In a subgroup analysis, iron-supplemented children who were anemic at baseline had had better outcomes of intelligence (SMD 0.79, 95%CI: 0.41, 1.16, P = 0.001) and memory (SMD 0.47, 95%CI: 0.13, 0.81; P = 0.006).

**Funding:** This work was conducted under the PhD studies of BTG, whose scholarship was partially funded by the Flemish Interuniversity Council (VLIR-UOS) in the context of the Institutional University Cooperation Program (IUC) with Arba Minch University https://www.vliruos.be/en/projects/project/22?pid=3604. The funders had no role in study design, data collection and analysis, decision to publish, or preparation of the manuscript.

**Competing interests:** The authors have declared that no competing interests exist.

**Abbreviations:** DHA, Docosahexaenoic acid; EPA, Eicosapentaenoic acid; LMICs, Low- and middle-income countries; RoB, Risk of bias; RoB2, Cochrane risk of bias assessment tool; SD, Standard deviation; SMD, Standardized mean difference.

## Conclusion

Iron supplementation has a significant positive effect on the intelligence, attention and concentration, and the memory of school-age children but there was no evidence on the effect of iron supplementation on their school achievement.

## Introduction

Anemia, defined as a condition in which the red blood cell oxygen-carrying capacity is insufficient to meet physiologic needs [1, 2], is the most prevalent nutritional problem globally, and more especially in low- and middle-income countries (LMICs) [3, 4]. Anemia can result from decreased erythrocyte production, increased loss of erythrocytes, or blood loss [2, 5]. Micronutrient deficiencies, infections and genetic disorders can all cause anemia through different mechanisms. Nutritional deficiency anemia results from a poor body status in iron, folic acid, vitamin B12, and/or vitamin A [2, 4, 6]. Anemia affects more school-age children [7, 8], at an age where their brain development continues. One in four school-age children suffers of anemia worldwide [9, 10].

Iron deficiency is the leading cause of anemia. Iron deficiency anemia accounts for nearly half of the anemia cases especially in LMICs [6, 10]. Iron is an essential element for heme synthesis, a precursor of hemoglobin, transitional storage of oxygen in tissues, and for transport of electrons through the respiratory chain. Iron also acts as a cofactor to various enzymes [11, 12]. Iron is an essential nutrient for the development and functioning of the brain. Its functions include ATP production, synthesis and packaging of neurotransmitters and uptake and degradation of neurotransmitters [6, 13, 14].

Studies have found that iron supplementation has an effect on some of the domains of cognitive development among school-age children [15–17]. In addition, the evidence is strong on the relationship between anemia and cognitive development [15, 18, 19]. Iron deficiency and iron deficiency anemia can cause cognitive deficits [20, 21].

Iron supplementation of school-age children is recommended in settings where anemia is prevalent but the evidence regarding its effectiveness on cognitive development is limited [22]. Iron supplementation improved hemoglobin concentration and reduced the incidence of anemia and iron deficiency in school age children [15, 22, 23]. Furthermore, iron supplementation was shown to be effective in improving cognition, safe in malaria settings, and had no gastrointestinal adverse effects [15, 16, 24]. A systematic review assessing the evidence of the effect of iron supplementation in school-age children's cognition dated to 2013 [15]. The aim of this systematic review and meta-analysis is to review the evidence from published randomized clinical trials to evaluate the effects of iron supplementation on cognitive development and function among school-age children.

## Methods and design

The Preferred Reporting Items for Systematic Reviews and Meta-Analyses (PRISMA) was used and performed for this systematic review and meta-analysis (S1 Checklist) [25]. The systematic review protocol was developed and registered on PROSPERO (CRD42021231262) before the literature search was performed on April 20, 2021 and updated on October 13, 2022.

### Data sources and search strategy

Five databases including MEDLINE (via the PubMed interface), EMBASE (via the embase.com interface), Scopus, Web of Science, and the Cochrane library (Cochrane Central Register

of Controlled Trials) were used to search for articles. The search strategy included terms on the following themes: cognition, school-age children, and iron supplementation. Studies eligible for this review involved randomized control trials or clinical trials. A detailed search protocol is provided in supplementary materials (S1–S5 Tables).

## Eligibility criteria

Studies were included if the study population was school-age children (age ranging between six to 12 years), the design was a randomized controlled trial, and they assessed iron supplementation effects on cognitive development indicators. There was no restriction regarding duration of the intervention, setting, language, or the year of publication. Studies conducted on children under five years only or over 12 years of age only, with chronic morbidities (e.g. HIV/AIDS), with developmental disability, or only on children exposed to known factors affecting cognitive development (e.g. lead, arsenic) were excluded. Children exposed to lead are at risk for decline in cognitive function and educational achievement [26–29]. Studies which assessed a treatment dose of iron for anemic children were also excluded. In addition, studies which did not assess the sole effect of iron supplementation on cognition were not included (for instance studies that assessed the effects of supplementation with multiple micro-nutrients with iron as one of the components with no control group that is supplemented with the same form of micronutrients excluding iron). Studies which provided iron together with folic acid or vitamin C, or provided deworming treatment were included. These were exceptionally accepted because iron is commonly provided with folic acid (a form supplemented during pregnancy), with vitamin C (as enhancer of iron absorption), and/or together with deworming because of the prevalence of intestinal parasites in LMICs.

## Outcome of the studies

The main outcome of the study intervention is the change in cognitive development between baseline and after supplementation. The outcome measurements were based on one of the cognitive domain measurements: intelligence, memory, attention, concentration, or academic achievement. The additional outcomes included anemia (hemoglobin concentrations below 115.0 g/l), hemoglobin concentrations (g/l), iron deficiency (measured using indicators of iron status including ferritin $<15$ μg/L and transferrin saturation $<15\%$) [5, 7, 8], iron deficiency anemia (coexistence of anemia and iron deficiency), safety outcomes (as indicated by the number of event morbidity and mortality during supplementation, diarrhea, constipation, vomiting and loss of appetite) and adherence to the supplementation.

## Study selection

Identified studies using the search strategy were exported from the respective databases to EndNote X9 (www.endnote.com). EndNote was used to identify and drop duplicates. Screening of records based on the title and abstract, and of full text was carried out using DistillerSR (https://www.evidencepartners.com/), a systematic review software. The first round screening conducted in April, 2021 was carried out by two reviewers (BTG, ND, MB, and GE), who independently included relevant articles following the above mentioned eligibility criteria. Disagreements were resolved by a third reviewer and consensus-based discussions. Screening of the updated records in October, 2022 was carried out by one reviewer (BTG) [30].

## Data extraction

Data extraction was carried out independently by two reviewers (BTG & ND). The format for the extraction of the data, based on the Cochrane Data collection form for intervention reviews (https://dplp.cochrane.org/data-extraction-forms), was modified based on the study question and pre-tested before use. The intervention consisted of supplementing iron compared to a control group that enables the comparison of the sole effect of iron. Details of the iron dose, form, frequency (intermittent, daily), duration of the intervention, and the provision method (at school/home, by teachers or caregivers) were considered. Data extraction form included the population description, study setting, inclusion and exclusion criteria, selection criteria, blinding, number of participants (cluster if applicable), and withdrawals from the study. Biochemical and development outcomes were reported as mean and standard deviations (SD), median or interquartile range and 95% confidence intervals (IQR, 95%CI). Disagreements were discussed to achieve a harmonized approach of data extraction.

## Assessment of risk of bias

The risk of bias (RoB) was assessed using the Cochrane risk of bias assessment tool (RoB2) (https://sites.google.com/site/riskofbiastool/welcome/rob-2-0-tool). Briefly, randomization, concealment of allocation prior assignment, blinding of participants, personnel, and outcome assessors; assessment of incomplete outcome data; assessment of selective reporting; and checking for possible attrition bias through withdrawals, dropouts, and protocol deviations were assessed. Two reviewers made judgments of bias independently. Disagreements were resolved by consulting with a third reviewer for arbitration as necessary.

## Data synthesis

Review Manager Software (Review Manager 5.4) was used for data analysis. Descriptive statistics (means and SD, median, IQR, 95%CI and proportions) were used to summarize baseline information.

Cognitive development outcomes were organized with respect to the cognitive domain. They were presented as pooled standard mean difference (SMD) with 95%CI. The pooled SMD was calculated using the random effect model. The approximation in case of missing mean (SD) was done based on median and IQR [31]. In case there were more than one group supplemented with iron (different doses or frequency of supplementation) or in control groups, the mean (SD) was pooled when there was no significant difference between the effects of the groups. If there was significant difference between the effect of the groups, the groups were presented separately according to Cochrane Review Manual [32].

A qualitative synthesis was performed in the case of studies where data were not sufficient to be included in the meta-analysis. The unit of analysis is the individual study participant in the case of randomized controlled trials. In cluster randomized controlled trials, clustering was also considered as recommended by the Cochrane Handbook for Systematic Reviews of Interventions. Heterogeneity was checked by Foster plot using Chi-square test and $I^2$ statistic, which describes the percentage of variability in effect estimate due to heterogeneity. In a sensitivity analysis, we 1) compared all combined iron supplemented groups with the control/placebo group, and 2) excluded studies that supplemented iron with folic acid or vitamin C. In addition, a sensitivity analysis was performed to investigate the overall effect as one study was excluded sequentially. Publication bias was assessed using Funnel plot asymmetry and Eggen test.

**Analysis of subgroups.** Heterogeneity was assessed for the following variables: duration of the follow-up (less than four months and four and more months of supplementation),

frequency of iron supplementation (four or more times per-week and once or twice per-week), baseline anemia and iron status, and study quality. Forest plots of the subgroup analyses are presented when the data were available.

## Results

In total, five databases were searched on April 20th, 2021. The data search resulted in a total of 6599 records after 124 duplicates dropped using EndNote. The screening based on the title and abstract, done in duplicate, resulted in 40 records that were screened based on full text. In total, 27 articles were excluded based on the eligibility criteria. On October 13[th], 2022, the search was run to retrieve new records published since the first search has been completed. After merging the old and the new EndNote libraries and dropping all the duplicates (i.e., records that were already in the old library), 1070 new records have been retrieved. The screening based on title and abstract resulted in one record [33], an abstract in a conference proceeding which was excluded for no full article has been published. Finally, thirteen articles were included in this systematic review (Fig 1).

### Study characteristics

Using the World Bank income-based country classification, two studies were conducted in high-income countries (Croatia and United Kingdom) [35, 36], ten in middle-income countries (one in South Africa, one in Egypt, two in Indonesia, two in Thailand, and four in India) [37–46] and one study was conducted in a low-income country (Mali) [47]. In the study conducted in India by Gopaldas and collaborators, only boys were enrolled [45], while only girls were enrolled in the studies conducted by Kashyap & Gopaldas (1987) and Sen & Kanani (2009) [38, 44]. One study considered anemic and infected children with *Schistosoma haematobium* [47], and one study considered iron deficiency as an inclusion criterion [37]. In nine studies, iron supplements were in the form of ferrous sulfate, in doses ranging from 2 to 60 mg of elementary iron [37, 39–45, 47]. Other forms of iron supplements were also used either in a singular form (i.e. ferri-glycine sulfate) [36] or in combination with folic acid, or vitamin C [35, 38, 46]. Two studies supplemented additionally long chain fatty acids (eicosapentaenoic acid (EPA) and docosahexaenoic acid (DHA)) [37], and multiple micronutrients in a factorial design [47]. Placebo control was not included in three studies [38, 46, 47] (Table 1).

In most of the studies supplements were provided at least five days a week. Children were supplemented once or twice weekly, four times per week or daily in three studies [37–39]. Five included studies supplemented children for 2–3 months. Five studies provided the supplement for 4 months and three studies for 8 months to a year.

Different types of cognitive development tests were used (Table 1). Based on cognitive domain categorization, 11 studies assessed intelligence in general, five studies assessed attention & concentration, and five studies assessed memory. In addition, we included school achievement as an outcome which was assessed in six studies.

### Risk of bias

Table 2 presents the summary of the RoB assessment of the included articles based on RoB2. Six studies were considered as high RoB [38, 40–42, 46, 47], four represent some concern [35, 36, 43, 44] and three have low RoB [37, 39, 45]. The study by Baumgartner *et al.* 2012 reported no significant dropout rate among the intervention groups. The frequency of supplementation was changed from 4 days per week to 5 days per week to catch up on the unexpected loss of intervention days due to the interruptions in the supplementation [37]. Sungthong *et al.* (2004) did not report the exact number of dropout rate by study group. However, we estimated

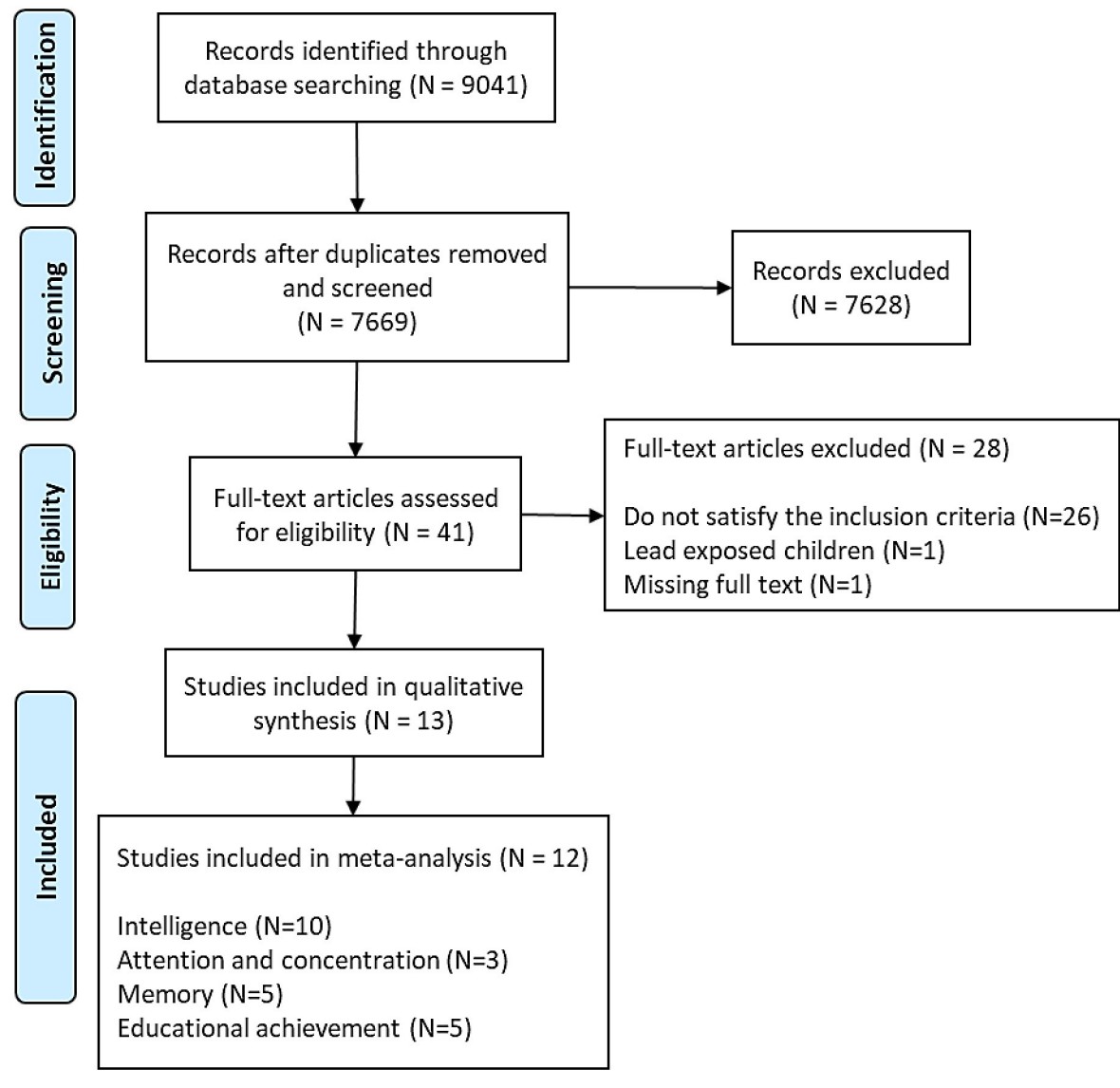

**Fig 1. PRISMA study flow diagram for the systematic review [34].**

that the number of dropout rate was higher in the placebo (≈28 children) than weekly iron supplemented group (≈20 children) and daily iron supplemented group (≈11 children). The analysis between the dropout children and the children who completed the study showed no significant difference [39]. Lynn and Harland (1998) and Soemantri (1989) did not report the randomization process of the children to the intervention groups [35, 41].

## Effect of iron supplementation on cognitive development of school-age children

Cognitive outcomes were categorized into three domains: intelligence, attention and concentration, and memory. In addition, educational achievement was also included.

**Effect of iron supplementation on general intelligence of the school-age children.** Eleven studies assessed the general intelligence of the children using five tools. The tests used

**Table 1. Descriptive summary of the studies included in the systematic review.**

| Country (Author, year) | Number of subjects, age, intervention duration | Baseline anemia/iron status | Intervention/ control and groups | Cognition outcome assessment method |
|---|---|---|---|---|
| Baumgartner *et al.* 2012 [37] (South Africa) | n = 319, age 6–11 y, 8.5month | Included Iron deficiency only Anemia prevalence: 20.9% | 50 mg iron as iron sulfate 420 mg DHA and 80 mg EPA Placebo tablets 4 groups: Iron & placebo; Placebo & DHA/ EPA; Iron & DHA/EPA; Placebo & placebo, Frequency 5 day per week | Kaufman Assessment Battery (Atlantis, Atlantis Delayed, Hand movement (**I**) and Triangles) & HVLT (Recognition, Discrimination index & Total scores of recalls (**M**)) |
| Sen & Kanani 2009 [38] (India) | N = 240 Female, 9–13 y, 1 year | Mean Hb 11.3g/dL Anemic 68.3% Unknown Iron status | 100 mg elemental iron and 0.5 mg folic acid (IFA) Control: no supplementation IFA once weekly; IFA-twice weekly; IFA-Daily & control | Clerical task (AC), Digit Span (M), Mazes test (I), & Visual Memory Test |
| Sungthong *et al.* 2004 [39] (Thailand) | N = 397, 6–13 y, 16 weeks | Anemic: 27% Iron deficient: 21.5% | 400 mg Albendazole for all, 300mg ferrous sulfate (60 mg elemental Fe); Placebo tablet Daily iron, weekly iron, & placebo | IQ point (TONI II) (I), Thai language, Mathematics (EA) |
| Soemantri 1989 [41] (Indonesia) | N = 130, 8.1–11.6y, 3 months | Iron deficiency Anemic: 58, Nonanemic: 72 | 10 mg per kg of body weight per day of ferrous sulfate (equals 2mg elemental Fe), Placebo: saccharin and tapioca Groups: iron suppl. &anemic; iron suppl. &non-anemic; placebo &anemic; placebo &non-anemic | IQ point (I), Language score, Mathematics score (EA), Biology scores, & Social science score |
| Pollitt *et al.* 1989 [42] (Thailand) | N = 1358, 9–11 y, 16 weeks | Iron deficiency anemia: 101 (7.44%); iron depleted: 47 (3.46%) | 200mg of albendazole for all, 50mg/day ferrous sulphate first 2 weeks (2 mg elemental Fe per kg of body weight per day) then 100mg/day (4 mg per kg of body weight per day) for 14 weeks, Placebo: sweet cassava powder | Thai Language, Mathematics (EA), Raven's Colour Progressive Metrices (I) |
| Kashyap & Gopaldas 1987 [44] (India) | N = 166 Female, 8–15 y, 8 months | Hb (g/dL) level of iron suppl.: 10.28; placebo: 10.39 | 60 mg elemental iron per day (FeSO$_4$), Placebo: sugar tablets | Clerical Task (AC) test, Digit Span test (M), Mazes test (I), Visual Memory test |
| Soemantri *et al.* 1985 [40] (Indonesia) | N = 119, Mean age 10.80 y, 3 months | Iron deficiency anemia status:78 Non-anemic: 41 | 10 mg per kg of body weight per day of ferrous sulfate (equal to 2 mg of elemental iron), Placebo: only saccharine and tapioca | Raven Progressive Matrices, The Bourden-Wisconsin test (AC), School achievement test (Mathematics, biology, social science, and language) |
| Gopaldas *et al.* 1985 [45] (India) | N = 48, Male only 8–15 y, 4 months | Mean (SD) Hb (g/dL) 11.37 (0.59) No difference between groups Iron status: Unknown | Groups: 40mg iron suppl.; 30mg iron suppl in the form of ferrous sulphate; placebo: brown sugar Frequency of supplementation: Daily | Visual Recall, Digit Span (M), Mazes (I), Clerical Task (AC) |
| Seshadri *et al.* 1982 [46] (India) | N = 94, 5–8 y, 60 days | Mean (SD) of baseline Hb (g/ L): 102.3(10.3) with no difference between groups Iron status: Unknown | Iron suppl.: 20 mg Fe & 0.1 mg folic acid; control: No intervention Frequency of supplementation: Daily | Verbal test (WISC), performance tests (WISC), total IQ (I) |
| Lynn & Harland 1998 [35] (England) | N = 413, 12–16 y, 16 weeks | Anemic = 2.9%, Iron deficiency = 16.9% | 17 mg elemental iron with 70 mg ascorbic acid; Placebo tablet (not specified) Frequency of supplementation: Daily | Raven's Colour Progressive Metrices (I) |
| Buzina-Suboticanec *et al.* 1998 [36] (Croatia) | N = 66, 8.7–9.6 y, 10 weeks | Mean (SD) Hb (g/L) 120.4 (5.2), Serum iron (mol/L) 17.4 (7.2) | 100 mg of iron in the form of ferri-glycine sulfate; placebo Frequency of supplementation: 6 days/week | Verbal (Arithmetic, Similarities, Digit span) & Non-verbal (Picture completion, Block design, Coding) |
| Ayoya *et al.* 2012 [47] (Mali) | N = 439, 7–12 y, 12 weeks | Included: Anemic and infected with S. haematobium Mean (SD) Hb (g/L) 104.9(10) | 60 mg ferrous sulfate; MM supplements All children treated with Praziquantel (40 mg per kg of body weight) Groups: Iron suppl. only; MM suppl. only; iron & MM suppl.; and control (no intervention) Frequency of supplementation: 5 days/week | School attendance, School achievement test (EA) |

*(Continued)*

**Table 1.** (Continued)

| Country (Author, year) | Number of subjects, age, intervention duration | Baseline anemia/iron status | Intervention/ control and groups | Cognition outcome assessment method |
|---|---|---|---|---|
| Pollitt 1997 [43] (Egypt) | N = 68, 8–11 y, 4 months | Iron deficiency anemic: 28; nonanemic: 40 | 50 mg ferrous sulfate (2 mg elemental iron per kg body weight); placebo: Unknown Frequency of supplementation: 6 days/week | Continuous Performance Test, Peabody Picture Vocabulary Test (PPVT), Matching Familiar Figure Test (MFFT). |

AC: attention and concentration; DHA: docosahexaenoic acid; EA: Educational achievement; EPA: Eicosapentaenoic acid; Hb: Hemoglobin concentration; I: Intelligence; IFA: Iron folic acid; M: Memory; WISC: Wechsler Intelligence Scale for Children; y:year

in the assessment are Kaufman Assessment Battery (Hand movement), Mazes test, test of Nonverbal Intelligence (TONI II), Raven's Color progressive matrices, and Wechsler Intelligence Scale [35–39, 41–46]. Five studies showed that iron supplementation increased significantly the intelligence of the children [36, 38, 44–46]. A study by Kashyap and Gopaldas (1987) indicated that the intelligence of the schoolgirls was significantly improved among iron supplemented groups (supplemented 60 mg elemental Fe/day for four months). The effect was sustained after suspending the supplementation for 4 months [44]. However, the study conducted by Soemantri (1989) showed no effect of daily 2 mg per kg of body weight, elemental iron supplementation provided in the form of iron sulfate during 3 months on intelligence of

**Table 2. Risk of bias among the included studies using RoB 2 tool.**

| First author, publication year and [reference] | Risk of bias domain | | | | | Overall risk of bias |
|---|---|---|---|---|---|---|
| | D1 | D2 | D3 | D4 | D5 | |
| Baumgartner *et al.* 2012 [37] | L | L | L | L | L | L |
| Sen & Kanani 2009 [38] | H | H | H | L | SC | H |
| Sungthong *et al.* 2004 [39] | L | L | L | L | L | L |
| Soemantri 1989 [41] | SC | SC | L | L | SC | H |
| Pollitt *et al.* 1989 [42] | SC | SC | SC | L | L | H |
| Kashyap & Gopaldas 1987 [44] | SC | L | L | L | L | SC |
| Soemantri *et al.* 1985 [40] | H | L | L | L | L | H |
| Gopaldas *et al.* 1985 [45] | L | L | L | L | L | L |
| Seshadri *et al.* 1982 [46] | SC | SC | L | SC | L | H |
| Lynn & Harland 1998 [35] | SC | L | L | L | L | SC |
| Buzina-Suboticanec *et al.* 1998 [36] | SC | L | L | L | L | SC |
| Ayoya *et al.* 2012 [47] | SC | SC | L | L | L | H |
| Pollitt 1997 [43] | SC | L | L | L | L | SC |

**Risk of bias domain**

D1: Randomization process

D3: Intervention Deviations

D3: Missing outcome data

D4: Measurement of the outcome

D5: Selection of the reported result

**Judgment**

L Low risk of bias

SC Some concern

H High risk of bias

the children [41]. Study by Gopaldas *et al.* (1985), showed that schoolboys who were supplemented with 40 mg iron for four months scored significantly higher in intelligence compared to the baseline and to those who had placebo supplements. Both 30 mg and 40 mg doses of iron improved significantly intelligence of anemic children [45]. Iron supplementation had significant effect on intelligence only among mildly anemic schoolchildren from Croatia [36]. Lynn and Harland (1998) demonstrated that iron supplementation did not affect significantly children's intelligence. But in subgroup analysis, supplementation of 17 mg of elemental iron combined with 70 mg of vitamin C for 16 weeks had significant effect on the intelligence of children with low iron status [35]. Sen and Kanani assessed the effect of once-weekly, twice-weekly, and daily iron-folic acid supplementation of schoolgirls with no intervention. Intelligence increased significantly compared to the non-supplemented group, but the effect was greater in the more-frequent iron-folic acid supplementation (daily and twice- weekly) than in the once-weekly supplemented group [38]. Paradoxically, the study conducted by Sungthong and colleagues found that improvement in intelligence of children was significantly lower in the daily iron supplemented children than in the group who received weekly iron supplementation or placebo [39].

Six studies reported no significant effect of iron supplementation on intelligence of school-age children [35, 37, 39, 41–43]. A study conducted by Pollitt (1997) in Egypt showed that iron sulfate supplementation of children aged 8–11 years had no significant effect on their intelligence [43].

**Effect of iron supplementation on general intelligence of the school-age children— Meta-analysis.** A total of 1703 and 1402 school-age children were assigned to iron supplement and placebo (or no intervention) groups in ten studies, respectively, included for meta-analysis [35–39, 41, 42, 44–46]. Overall, iron supplementation had significant effect on intelligence of school-age children (SMD 0.46, 95%CI: 0.19, 0.73, P<0.001, n = 3105; test for heterogeneity $I^2$ = 89%, P<0.001) (Fig 2). Iron supplementation improved significantly the intelligence in studies categorized with high RoB (SMD 0.77, 95%CI: 0.11, 1.43, P = 0.02, n = 1744; test for heterogeneity $I^2$ = 94%, P<0.01). There was no significant effect in studies with low and moderate RoB (SMD 0.25, 95%CI: -0.05, 0.54; P = 0.10, n = 1361; test for heterogeneity $I^2$ = 82%, P<0.01) (S1 Fig). There was no significant effect of iron supplementation on the intelligence after the studies which supplemented iron with folic acid [38, 46] or vitamin C [35] were excluded (SMD 0.20, 95%CI: -0.04, 0.44; P = 0.10, n = 2434; test for heterogeneity $I^2$ = 81%, P<0.01) (S2 Fig).

Four studies supplemented the children for less than four months and six studies for longer periods. The results showed that iron supplementation for shorter periods improved significantly the intelligence of school-age children (SMD 0.93, 95%CI: 0.12, 1.75; P = 0.03, n = 335; test for heterogeneity $I^2$ = 90%, P<0.01) (Fig 2). The effect remained significant after the study supplementing iron-folic acid [46] was excluded (S2 Fig). The effect was not significant when children were supplemented for longer periods (for four and more months) (SMD 0.21, 95% CI: -0.02, 0.43, P = 0.07, n = 2770; test for heterogeneity $I^2$ = 82%, P<0.01) (Fig 2). The sensitivity analysis showed similar finding after excluding the studies supplementing iron with folic acid [38] or vitamin C [35] (S2 Fig).

Iron supplementation for four and more times per week (including daily supplementation) has significant effect on intelligence (SMD 0.48, 95%CI: 0.17, 0.79, P = 0.003, n = 2806; test for heterogeneity $I^2$ = 91%, P<0.01) (Fig 2). However, there was no significant effect of iron supplementation for four and more times per week (SMD 0.24, 95%CI: -0.03, 0.52, P = 0.08, n = 2243; test for heterogeneity $I^2$ = 83%, P<0.01) on intelligence after excluding studies that supplemented iron with folic acid [38, 46] or vitamin C [35] (S2 Fig). Weekly iron supplementation does not affect significantly the intelligence of school-age children (SMD 0.18, 95%CI:

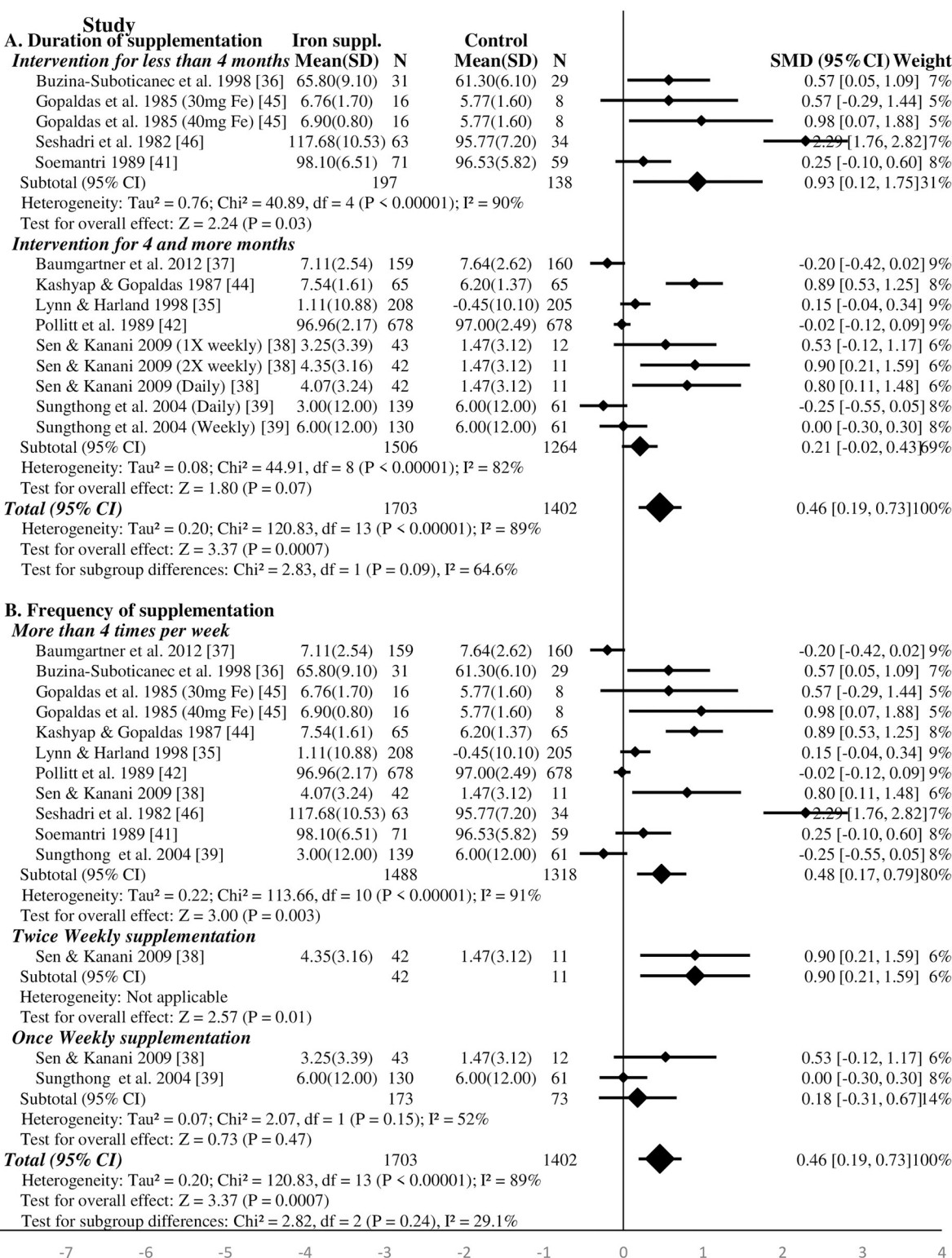

**Fig 2. Forest plot, SMD analysis of the effect of iron supplementation on intelligence of school-age children (subgroup analysis for duration of intervention and frequency of supplementation).**

-0.31, 0.67, P = 0.47, n = 246; test for heterogeneity $I^2$ = 52%, P = 0.15) (Fig 2). There was not sufficient studies to carry out the sensitivity analysis regarding the effect of the sole weekly supplementation of iron.

In a subgroup analysis, iron supplementation significantly improved the intelligence of anemic children (SMD 0.79, 95%CI: 0.41, 1.16, P = 0.001, n = 236; test for heterogeneity $I^2$ = 40%, P = 0.17) (Fig 3). The effect of iron supplementation was not significant in non-anemic (SMD 0.01, 95%CI: -0.10, 0.12, P = 0.89, n = 1295; test for heterogeneity $I^2$ = 0%, P = 0.73), iron deficient (SMD -0.45, 95%CI: -2.07, 1.17, P = 0.59, n = 174; test for heterogeneity $I^2$ = 96%, P<0.01), and iron replete (SMD 0.09, 95%CI: -0.10, 0.29, P = 0.35, n = 1655; test for heterogeneity $I^2$ = 90%, P<0.01) school-age children at baseline (Fig 3). The effect of iron supplementation solely (excluding iron-vitamin C [35]) on intelligence among iron deficient and iron replete school-age children at baseline were similar (S2 Fig). The sensitivity analysis showed that iron supplementation significantly improved the intelligence of iron deficient children (SMD 0.52, 95%CI: 0.16, 0.87, P = 0.005, n = 128; test for heterogeneity $I^2$ = 0%, P = 0.77) after dropping the study of Pollitt *et al.* (1989) [42].

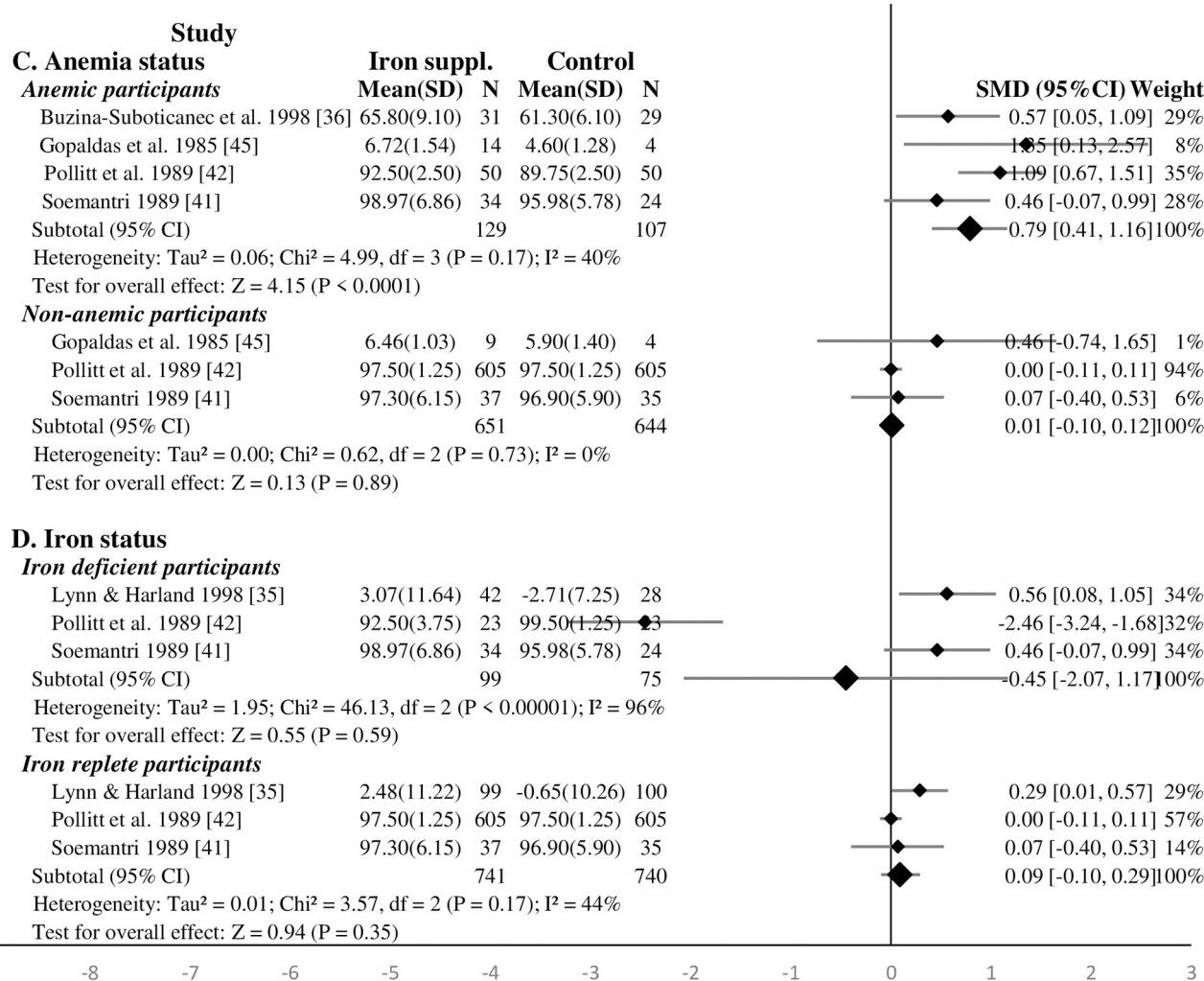

**Fig 3. Forest plot, SMD analysis of the effect of iron supplementation on intelligence of school-age children (subgroup analysis for anemia and iron status).**

**Effect of iron supplementation on attention and concentration of school-age children.** Five studies assessed the effect of iron supplementation on attention and concentration of school-age children using Clerical Task, the Bourden-Wisconsin test, and Continuous Performance Test [38, 40, 43–45]. Four studies found that iron supplementation had significant improved attention and concentration of school-age children [38, 40, 43–45]. In a yearlong interventional study by Sen & Kanani (2009) indicated that frequent iron-folic acid supplementation (daily or twice weekly) was found to improve significantly attention and concentration compared to less frequent supplementation (once weekly) [38]. Soemantri and colleagues reported a significant effect of iron supplementation in all the children. They also showed that the improvement in attention and concentration was significant among iron deficient anemic children but not in those who were non-anemic [40]. Similarly, the study conducted by Pollitt (1997) demonstrated that iron supplementation improved significantly the attention and concentration of anemic but not non-anemic school-age children [43].

**Effect of iron supplementation on attention and concentration of school-age children—Meta-analysis.** Only three out of five studies were included in the meta-analysis. In total, 224 and 115 children were respectively supplemented with iron and placebo (or no supplement) [38, 44, 45]. Overall, iron supplementation improved significantly the attention and concentration of school-age children (SMD 0.45, 95%CI: 0.09, 0.80, P = 0.01, n = 339; test for heterogeneity $I^2$ = 49%, P = 0.10). Assessed for the RoB, one study had a high RoB while two had low and moderate RoB. When the high RoB study was excluded from the analysis, iron supplementation had no significant effect on the attention and concentration of school-age children (SMD 0.33, 95%CI: -0.39, 1.05, P = 0.37, n = 178; test for heterogeneity $I^2$ = 77%, P = 0.04) (S3 Fig). Similarly, iron supplementation had no significant effect on the attention and concentration of school-age children after excluding the study that supplemented iron-folic acid [38] (SMD 0.33, 95%CI: -0.39, 1.05, P = 0.37, n = 178; test for heterogeneity $I^2$ = 77%, P = 0.04) (S4 Fig).

Iron supplementation for short duration (less than four months, two studies [44, 45]) had no significant effect on the attention and concentration of school-age children (SMD 0.33, 95%CI: -0.39, 1.05, P = 0.37, n = 178; test for heterogeneity $I^2$ = 77%, P = 0.04). Similarly, frequent iron supplementation (four or more times per-week) had no significant effect on the attention and concentration of children (SMD 0.41, 95%CI: -0.04, 0.86, P = 0.07, n = 231; test for heterogeneity $I^2$ = 54%, P = 0.11) (Fig 4). The effect remained non-significant after excluding the study with iron-folic acid supplementation [38] (S4 Fig). There was no evidence on effects of supplementation of iron for long duration (four or more months) or intermittently.

**Effect of iron supplementation on the memory of school-age children.** Five studies reported the effect of iron supplementation of school-age children on their memory using Digit Span, Visual Memory Test and Hopkins Verbal Learning Test tools [36–38, 44, 45]. Iron supplementation was found to have significant effect on the memory of the children in four out of five studies [37, 38, 44, 45]. The study by Baumgartner and colleagues assessed the sole and combined effects of supplementing iron and long chain fatty-acids (EPA/DHA) compared to placebo on the memory of school children. Iron supplementation improved significantly children's memory compared to children who received placebo. The effect of supplementation on memory of the school-age children was significantly higher among iron supplemented girls compared to boys and in anemic compared to non-anemic children [37]. Sen & Kanani reported that iron-folic acid supplementation increased memory in schoolgirls in India. Improvement in memory was greater in the frequently iron-folic acid supplemented groups, that is daily and twice weekly, than in the once weekly supplemented group [38]. Another study in India by Gopaldas and colleagues among boys showed that the mean test score for digit span increased significantly from 3.43 to 4.42 in the two iron supplemented

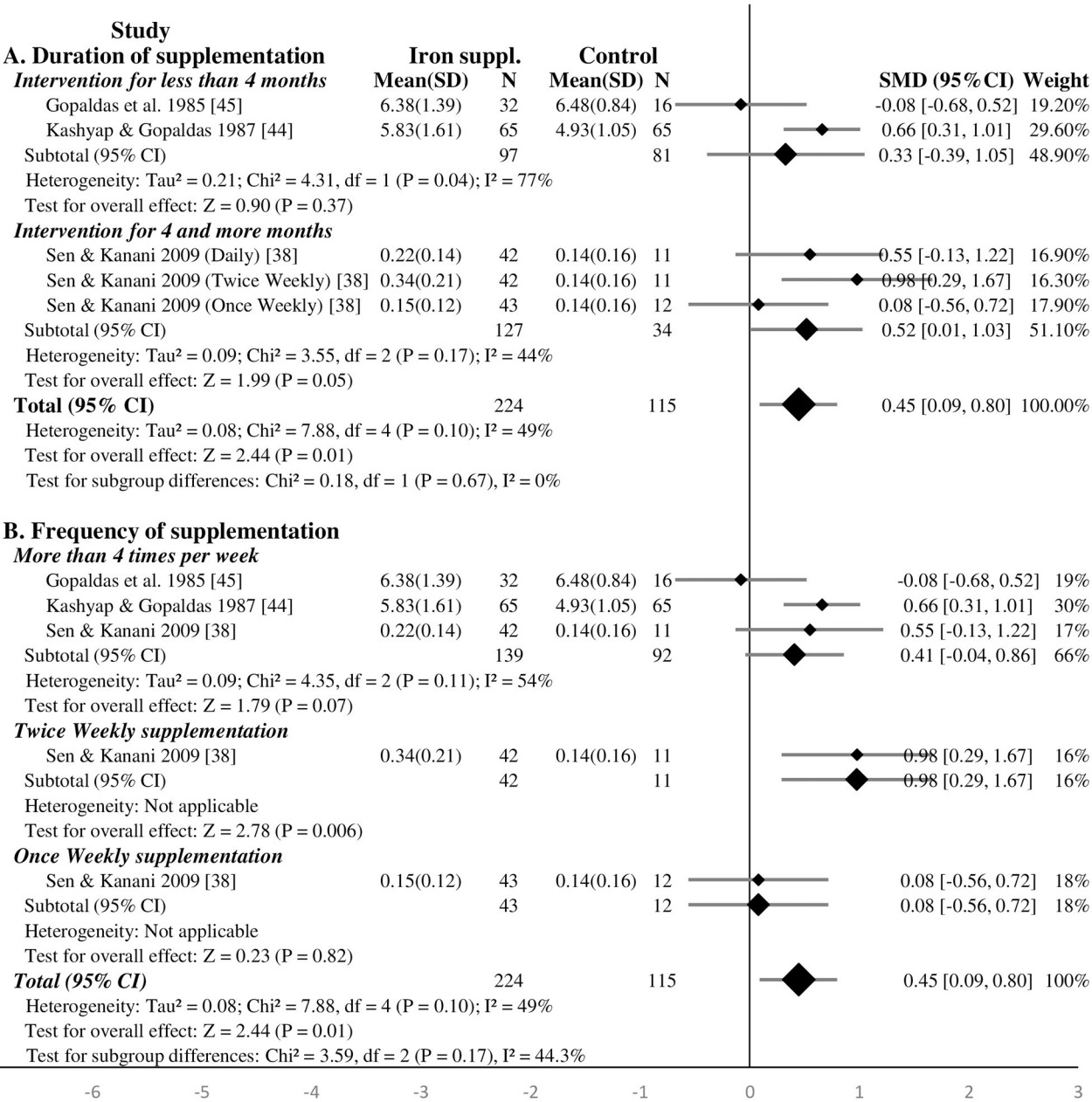

**Fig 4. Forest plot, SMD analysis of the effect of iron supplementation on attention and concentration of school-age children.**

groups (from 3.40 to 4.21 in children supplemented with 30mg iron and from 3.46 to 4.62 in those supplemented with 40mg iron), while there was no significant change in the placebo group (from 3.06 to 3.16). Significant improvement in visual recall test but not in digit span test compared to the placebo were found among the 30 mg iron supplemented group. In the 40 mg iron supplemented group, both digit span and visual recall tests were significantly higher compared to the placebo group [45]. The study conducted by Buzina-Suboticanec *et al.* (1998) did not find any effect in digit span after 10 weeks of iron supplementation compared to placebo [36].

**Effect of iron supplementation on the memory of school-age children—Meta-analysis.** In the five studies, a total of 414 and 304 children were assigned to receive iron supplementation and placebo (or non-intervention), respectively [36–38, 44, 45]. The meta-analysis showed that iron supplementation significantly improved the memory of school-age children (SMD 0.45, 95%CI: 0.04, 0.69, P <0.001, n = 718; test for heterogeneity $I^2$ = 45%, P = 0.09). After excluding the study with high RoB [38], the effect of iron supplementation on improving the memory of school-age children remained significant (SMD 0.30, 95%CI: 0.09, 0.51, P = 0.004, n = 557; test for heterogeneity $I^2$ = 18%, P = 0.30) (S5 Fig). The effect of iron supplementation on improving the memory of school-age children remained significant after excluding iron-folic acid supplementation study [38] (SMD 0.33, 95%CI: 0.08, 0.57, P = 0.008, n = 581; test for heterogeneity $I^2$ = 38%, P = 0.18) (S6 Fig).

The subgroup analysis showed that frequent iron supplementation (four or more times per-week) had significant effect on the memory of school-age children (SMD 0.45, 95%CI: 0.15, 0.74, P = 0.003, n = 610; test for heterogeneity $I^2$ = 58%, P = 0.05) (Fig 5). Similarly, the effect remained significant after the exclusion of iron-folic acid supplementation study [38] (S6 Fig). There was no evidence regarding twice or once weekly iron supplementation (Fig 5). The three studies which reported the effect of iron supplementation based on child's anemic status found that iron supplementation had significant effect on the memory of anemic children (SMD 0.47, 95%CI: 0.13, 0.81; P = 0.006, n = 145; test for heterogeneity $I^2$ = 0%, P = 0.99) but not on non-anemics (SMD -0.02, 95%CI: -1.01, 0.97; P<0.97, n = 266; test for heterogeneity $I^2$ = 65%, P = 0.09) (Fig 5).

**Effect of iron supplementation on educational achievement and school performance of school-age children.** Educational achievement was assessed by six studies using school attendance, overall grade, tests related to mathematics, language, biology, and social science courses [36, 39–42, 47]. Iron supplementation significantly improved the school performance of children in two studies [40, 41]. According to the study conducted by Soemantri (1989), iron supplementation had significant effect on math's score but not on language score in all children. In a subgroup analysis, educational achievement in both math's and language scores improved significantly in anemic children [41]. These results are in line with the study conducted by Soemantri *et al.* (1985), which found that improvement in educational achievement was greater in anemic children than in non-anemics [40]. Even if the effect of iron supplementation on educational achievement in a study conducted by Ayoya *et al.* (2012) in Bamako Mali was not significant, it showed a borderline effect with p-value of 0.08 [47].

**Effect of iron supplementation on educational achievement and school performance of school-age children—Meta-analysis.** Five studies were included in the meta-analysis [36, 39, 41, 42, 47]. One thousand two hundred nine (1209) and 1063 school-age children received iron supplementation and placebo/non-intervention, respectively. Iron supplementation did not have a significant effect on school achievement of the children (SMD 0.00, 95%CI: -0.21, 0.21, P = 1.00, n = 2272; test for heterogeneity $I^2$ = 76%, P<0.001) (Fig 6). Similar results were found when only two studies with low and moderate RoB were included in the analysis (SMD -0.16, 95%CI: -0.38, 0.05, P = 0.13, n = 382; test for heterogeneity $I^2$ = 0%, P = 0.65) (S7 Fig). In a subgroup analysis, school achievement of children was not affected by the child anemic status, nor by the duration of iron supplementation (Fig 6).

## Effect of iron supplementation on nutritional outcomes

Eleven studies [35–38, 41, 44–46, 48–50] out of the thirteen included in this systematic review reported on the effect of iron supplementation on hemoglobin concentrations of school-age children. Iron supplementation significantly increased the hemoglobin concentrations of the

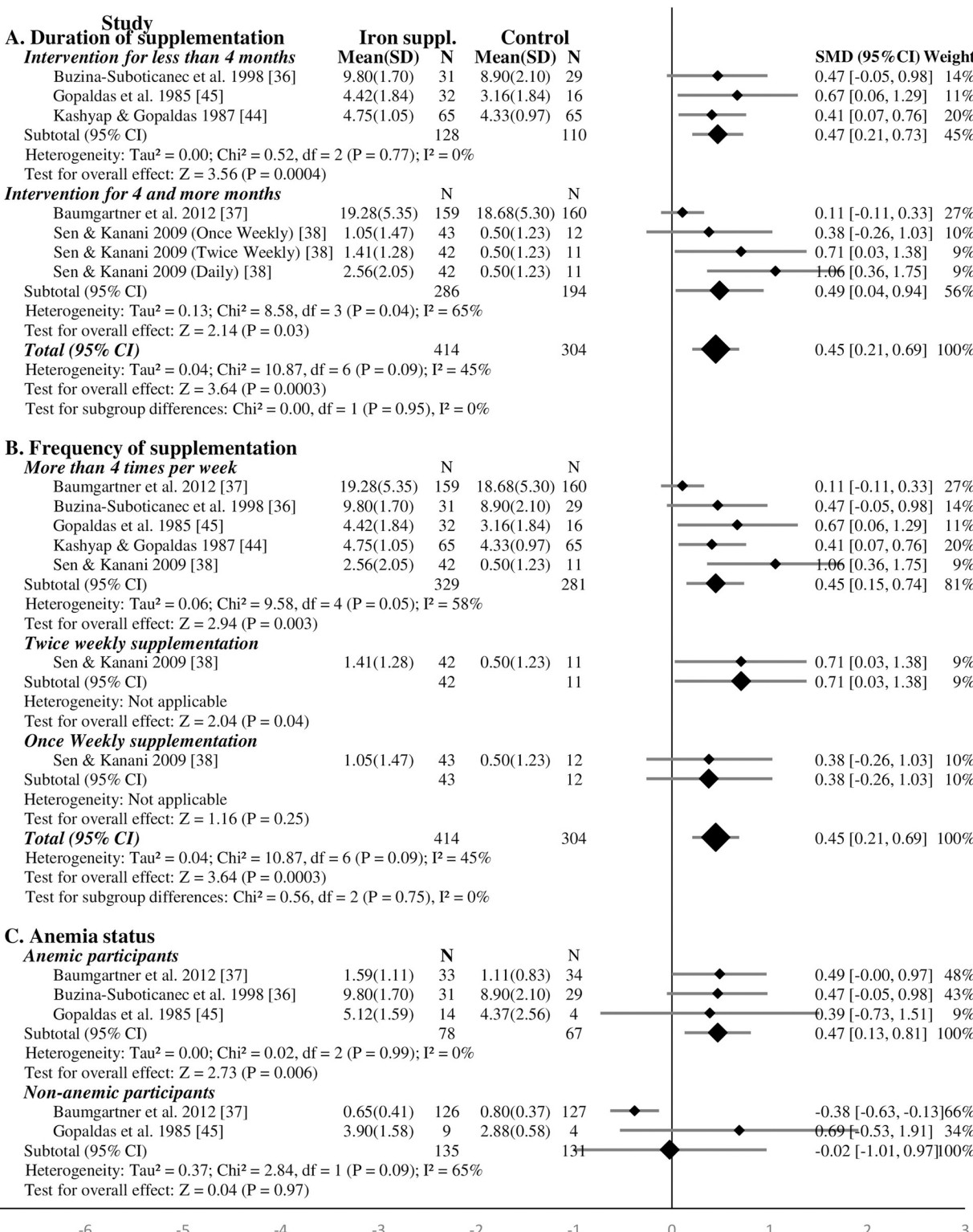

**A. Duration of supplementation**

| Study | Iron suppl. Mean(SD) | N | Control Mean(SD) | N | | SMD (95%CI) | Weight |
|---|---|---|---|---|---|---|---|
| *Intervention for less than 4 months* | | | | | | | |
| Buzina-Suboticanec et al. 1998 [36] | 9.80(1.70) | 31 | 8.90(2.10) | 29 | | 0.47 [-0.05, 0.98] | 14% |
| Gopaldas et al. 1985 [45] | 4.42(1.84) | 32 | 3.16(1.84) | 16 | | 0.67 [0.06, 1.29] | 11% |
| Kashyap & Gopaldas 1987 [44] | 4.75(1.05) | 65 | 4.33(0.97) | 65 | | 0.41 [0.07, 0.76] | 20% |
| Subtotal (95% CI) | | 128 | | 110 | | 0.47 [0.21, 0.73] | 45% |

Heterogeneity: Tau² = 0.00; Chi² = 0.52, df = 2 (P = 0.77); I² = 0%
Test for overall effect: Z = 3.56 (P = 0.0004)

| | | | | | | | |
|---|---|---|---|---|---|---|---|
| *Intervention for 4 and more months* | | N | | N | | | |
| Baumgartner et al. 2012 [37] | 19.28(5.35) | 159 | 18.68(5.30) | 160 | | 0.11 [-0.11, 0.33] | 27% |
| Sen & Kanani 2009 (Once Weekly) [38] | 1.05(1.47) | 43 | 0.50(1.23) | 12 | | 0.38 [-0.26, 1.03] | 10% |
| Sen & Kanani 2009 (Twice Weekly) [38] | 1.41(1.28) | 42 | 0.50(1.23) | 11 | | 0.71 [0.03, 1.38] | 9% |
| Sen & Kanani 2009 (Daily) [38] | 2.56(2.05) | 42 | 0.50(1.23) | 11 | | 1.06 [0.36, 1.75] | 9% |
| Subtotal (95% CI) | | 286 | | 194 | | 0.49 [0.04, 0.94] | 56% |

Heterogeneity: Tau² = 0.13; Chi² = 8.58, df = 3 (P = 0.04); I² = 65%
Test for overall effect: Z = 2.14 (P = 0.03)

| | | | | | | | |
|---|---|---|---|---|---|---|---|
| *Total (95% CI)* | | 414 | | 304 | | 0.45 [0.21, 0.69] | 100% |

Heterogeneity: Tau² = 0.04; Chi² = 10.87, df = 6 (P = 0.09); I² = 45%
Test for overall effect: Z = 3.64 (P = 0.0003)
Test for subgroup differences: Chi² = 0.00, df = 1 (P = 0.95), I² = 0%

**B. Frequency of supplementation**

| | | | | | | | |
|---|---|---|---|---|---|---|---|
| *More than 4 times per week* | | N | | N | | | |
| Baumgartner et al. 2012 [37] | 19.28(5.35) | 159 | 18.68(5.30) | 160 | | 0.11 [-0.11, 0.33] | 27% |
| Buzina-Suboticanec et al. 1998 [36] | 9.80(1.70) | 31 | 8.90(2.10) | 29 | | 0.47 [-0.05, 0.98] | 14% |
| Gopaldas et al. 1985 [45] | 4.42(1.84) | 32 | 3.16(1.84) | 16 | | 0.67 [0.06, 1.29] | 11% |
| Kashyap & Gopaldas 1987 [44] | 4.75(1.05) | 65 | 4.33(0.97) | 65 | | 0.41 [0.07, 0.76] | 20% |
| Sen & Kanani 2009 [38] | 2.56(2.05) | 42 | 0.50(1.23) | 11 | | 1.06 [0.36, 1.75] | 9% |
| Subtotal (95% CI) | | 329 | | 281 | | 0.45 [0.15, 0.74] | 81% |

Heterogeneity: Tau² = 0.06; Chi² = 9.58, df = 4 (P = 0.05); I² = 58%
Test for overall effect: Z = 2.94 (P = 0.003)

| | | | | | | | |
|---|---|---|---|---|---|---|---|
| *Twice weekly supplementation* | | | | | | | |
| Sen & Kanani 2009 [38] | 1.41(1.28) | 42 | 0.50(1.23) | 11 | | 0.71 [0.03, 1.38] | 9% |
| Subtotal (95% CI) | | 42 | | 11 | | 0.71 [0.03, 1.38] | 9% |

Heterogeneity: Not applicable
Test for overall effect: Z = 2.04 (P = 0.04)

| | | | | | | | |
|---|---|---|---|---|---|---|---|
| *Once Weekly supplementation* | | | | | | | |
| Sen & Kanani 2009 [38] | 1.05(1.47) | 43 | 0.50(1.23) | 12 | | 0.38 [-0.26, 1.03] | 10% |
| Subtotal (95% CI) | | 43 | | 12 | | 0.38 [-0.26, 1.03] | 10% |

Heterogeneity: Not applicable
Test for overall effect: Z = 1.16 (P = 0.25)

| | | | | | | | |
|---|---|---|---|---|---|---|---|
| *Total (95% CI)* | | 414 | | 304 | | 0.45 [0.21, 0.69] | 100% |

Heterogeneity: Tau² = 0.04; Chi² = 10.87, df = 6 (P = 0.09); I² = 45%
Test for overall effect: Z = 3.64 (P = 0.0003)
Test for subgroup differences: Chi² = 0.56, df = 2 (P = 0.75), I² = 0%

**C. Anemia status**

| | | | | | | | |
|---|---|---|---|---|---|---|---|
| *Anemic participants* | | N | | N | | | |
| Baumgartner et al. 2012 [37] | 1.59(1.11) | 33 | 1.11(0.83) | 34 | | 0.49 [-0.00, 0.97] | 48% |
| Buzina-Suboticanec et al. 1998 [36] | 9.80(1.70) | 31 | 8.90(2.10) | 29 | | 0.47 [-0.05, 0.98] | 43% |
| Gopaldas et al. 1985 [45] | 5.12(1.59) | 14 | 4.37(2.56) | 4 | | 0.39 [-0.73, 1.51] | 9% |
| Subtotal (95% CI) | | 78 | | 67 | | 0.47 [0.13, 0.81] | 100% |

Heterogeneity: Tau² = 0.00; Chi² = 0.02, df = 2 (P = 0.99); I² = 0%
Test for overall effect: Z = 2.73 (P = 0.006)

| | | | | | | | |
|---|---|---|---|---|---|---|---|
| *Non-anemic participants* | | | | | | | |
| Baumgartner et al. 2012 [37] | 0.65(0.41) | 126 | 0.80(0.37) | 127 | | -0.38 [-0.63, -0.13] | 66% |
| Gopaldas et al. 1985 [45] | 3.90(1.58) | 9 | 2.88(0.58) | 4 | | 0.69 [-0.53, 1.91] | 34% |
| Subtotal (95% CI) | | 135 | | 131 | | -0.02 [-1.01, 0.97] | 100% |

Heterogeneity: Tau² = 0.37; Chi² = 2.84, df = 1 (P = 0.09); I² = 65%
Test for overall effect: Z = 0.04 (P = 0.97)

**Fig 5. Forest plot, SMD analysis of the effect of iron supplementation on memory of school-age children.**

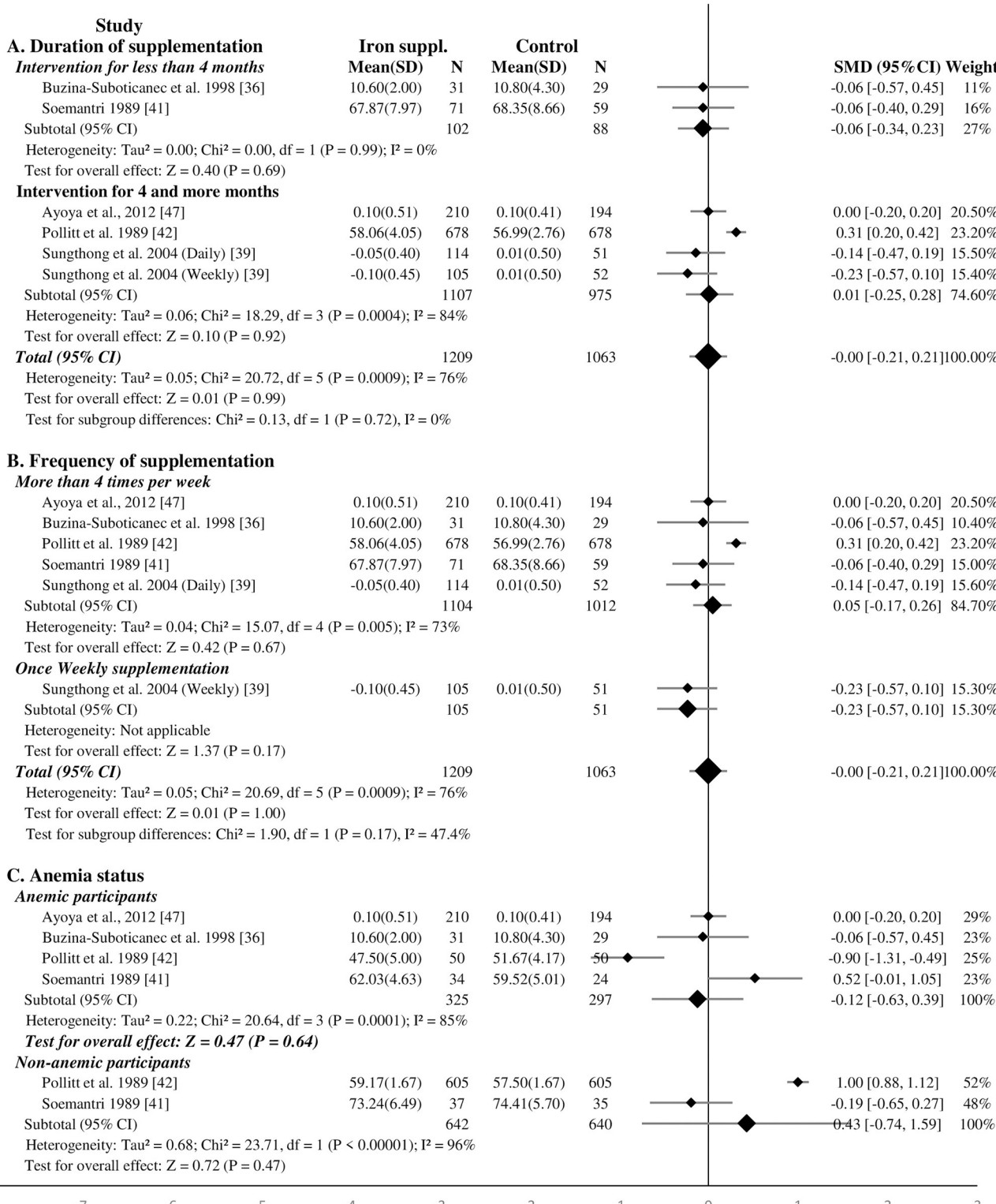

**Fig 6. Forest plot, SMD analysis of the effect of iron supplementation on school achievement of school-age children.**

children (SMD 1.08, 95%CI: 0.68, 1.49, P<0.001, n = 3613; test for heterogeneity $I^2$ = 96%, P<0.001) (S8 Fig). In a separate analysis that assessed the effect of iron supplementation on hemoglobin concentration categorized per the domain of the cognitive outcome, hemoglobin concentrations increased significantly in all categories compared to the placebo/non-intervention groups (S9 Fig). Only four studies reported on the effect of iron supplementation on serum ferritin [35, 48–50] and serum transferrin [36, 41, 48, 50]. The analysis indicated that iron supplementation did not have a significant effect on serum ferritin (SMD 1.93, 95%CI: -0.27, 4.14, P = 0.08, n = 2570; test for heterogeneity $I^2$ = 100%, P<0.01) and transferrin concentrations (SMD 1.01, 95%CI: -0.50, 2.51, P = 0.19, n = 1970; test for heterogeneity $I^2$ = 99%, P<0.01) (S10 & S11 Figs).

## Risk of iron supplementation on morbidity—Safety outcomes

Morbidity was reported by two studies. Malan and colleagues reported that iron supplementation, provided alone, significantly increased the number of days of illness (mostly respiratory), duration of episode of illness and the number of episodes [51]. Contrarily, Chwang and collaborators (1988) indicated that iron supplemented children had significantly lower morbidity scores, especially among those who were anemic [52]. Three studies were conducted in malaria-free areas [42, 49, 51]. Ayoya et al. (2009) indicated that malaria incidence and parasite density were higher in the iron supplemented group [48].

## Adherence to iron supplementation

Adherence to supplementation was reported in four studies. Overall adherence was reportedly high in the studies conducted by Baumgartner and colleagues (≃95%) [51] and Sungthong and colleagues (more than 93%) with no significant difference among intervention groups [37, 39, 49]. Sen & Kanani (2009) reported a mean adherence of 72% in the iron supplemented group (≥70% iron dose consumed in > 50% of participants), while the control group received no supplementation [38]. Soemantri (1989) included in the analyses only participants whose adherence was greater than 80% [41].

## Discussion

Thirteen randomized trials that evaluated the effect of iron supplementation on cognition of school-age children were included in the systematic review and twelve were included in the meta-analysis. Most of the studies included in this systematic review and meta-analysis were conducted in middle income countries. Only one study was conducted in low-income countries [47], despite the high prevalence of iron deficiency anemia in this setting [53].

Five out of the eleven studies which evaluated general intelligence showed that iron supplementation improved school-age children's intelligence. Eight out of ten studies included in a review by Lam & Lawlis (2017) on the effect of different micronutrients intervention on cognitive performance among school-age children indicated significant improvement in nonverbal fluid intelligence [18]. This meta-analysis showed that iron supplementation improved intelligence of school-age children, with moderate effect size. However, the effect on improved intelligence resulted only from four studies classified as high RoB [38, 41, 42, 46] and three studies classified as moderate RoB [35, 36, 44]. There was no evidence of effect of iron supplementation on the intelligence of school-age children in studies with low RoB [37, 39, 45]. The effect of iron supplementation on the intelligence of school-age children was not significant after excluding studies that supplemented children with iron with folic acid or vitamin C [35, 38, 46]. Studies showed that both folic acid and vitamin C (in animal studies) influence functionality of the brain. For instance, folic acid deficiency may result in the raise of homocysteine

concentration which is associated with poor cognitive performance [54–56]. Animal studies have shown that vitamin C affects neurodevelopment by influencing neuronal differentiation and myelin formation, and plays a role in neurotransmitters function [57, 58]. The study by Sungthong and colleagues reported that the general intelligence was lower among children who were supplemented daily than those in groups receiving weekly iron supplementation or placebo [39]. This might be due to the oxidative stress caused by high iron stores on cognitive function [59, 60], in case of iron replete status (6% iron deficiency anemia) and non-anemia (26% anemia) [39, 49].

Short periods (less than four months) of iron supplementation improved significantly intelligence of school-age children. Frequent administration (daily) of iron supplementation also improved intelligence of school-age children. However, our findings showed that frequent supplementation of iron, solely, has no effect on the intelligence of school-age children [36, 41, 45]. Additionally, iron supplementation had a greater significant effect on the intelligence of children who were anemic at baseline. Similar findings were reported in two previous meta-analyses [15, 16]. This effect was not found in children who were non-anemic and iron replete at baseline (Fig 3). Similarly, there was no effect of iron supplementation on the intelligence of children who were iron deficient at baseline. However this result was mainly driven by the findings of Pollitt *et al.* (1989), where iron-depleted children assigned to placebo, scored higher in Raven's Colour Progressive Metrices test (IQ) at baseline compared to iron-depleted children who received the iron treatment [42].

Iron supplementation improved attention and concentration of school-age children with moderate effect. Iron deficiency is associated with altered dopamine metabolism; dopamine is involved in regulating attention [61, 62]. The effect of iron supplementation was not significant after excluding the study with high RoB, which is also the study that supplemented children with iron- folic acid [38]. A previous review conducted on older children (6–18 years) and adults indicated that iron supplementation had statistically significant beneficial effects on attention or concentration [16]. Folic acid deficiency is associated with the reduction in the synthesis of S-adenosylmethionine, which inhibits S-adenosylmethionine-dependent methylenation reactions and neurotransmitters including dopamine [56].

Iron supplementation had a moderate effect in improving the memory of school-age children [37, 38, 44, 45]. This result remained significant when including all the studies, and after excluding the one study with high RoB [38]. In addition, the meta-analysis found that iron supplementation effects on memory was greater among anemic children. A systematic review and meta-analysis of iron supplementation on cognition of older children (6-18years) and adult found no evidence supporting that iron supplementation improves memory [16]. In animal studies, neuronal iron uptake is essential for normal hippocampal neuronal development and memory behavior [63, 64].

From the six studies which assessed the effect of iron supplementation on educational achievement of school-age children, only two showed a benefit of the interventions. This meta-analysis found no evidence that iron supplementation, regardless of its frequency and duration, and baseline anemic status of the children, had an effect on the educational achievement of school-age children. Similar results were reported in the review conducted by Falkingham *et al.* (2010), which indicated that there was no suggestion of the effect of iron supplementation on scholastic ability of children, regardless of their baseline anemic status [16]. The lack of effect of iron supplementation on educational achievement and school performance in children could be due to the longer time required to affect the domains of cognitive development compared to the time required to restore iron status [65, 66].

## Concerns regarding possible side effects

Iron supplementation can cause gastrointestinal side effects including loose stool/diarrhea, hard stool/constipation, abdominal pain, nausea, change in stool color, reflux/heartburn and headache [67]. Low doses of iron supplementation are reportedly not associated with an increased risk of infectious illness in children, except for a slight increase in the risk of developing diarrhea. Iron supplementation may increase the risk of malaria [60], and the WHO recommends monitoring fever in preschool and school children receiving intermittent iron supplementation in malaria settings [22]. Despite the serious risk that is evidenced with iron supplementation, side effects were reported in two studies only [51, 52], while the increased risk of malaria infection was reported in one study only [48] and three studies indicated that the setting was malaria-free [42, 49, 51]. It is important that morbidity and potential side effects are assessed in studies and programs that supplement iron to children.

## Strengths and weaknesses of the systematic review and meta-analysis

This systematic review updated the current evidence on the effects of iron supplementation on cognitive development and school achievement of school-age children. The review and the analyses were performed to report separately on the effects based on cognitive development domain of the children. In addition, school achievement was reported as a separate outcome.

Our study has limitations that are inherent to cognition as an outcome in research trials. The analysis was performed per domain of cognition. However, different tools were used to evaluate a specific domain which makes the comparison between the interventions' outcomes difficult. In addition, there were large differences in the intervention design including, the dose and the form of the iron supplements, and the frequency and duration of the supplementation. An optimal dose, frequency or duration of iron that results in improved cognitive development outcomes could not be established. Furthermore, the duration of the intervention to detect a change in cognitive development outcomes should be reconsidered. In addition, the role of other micronutrients in some studies, that might have affected the results could not be disentangled. Subgroup analysis was done based on overall RoB of the studies, frequency and duration of intervention, and anemia and iron status at baseline to explore the benefit per risk group. Because of the limited number of studies which reported on the micronutrient status of children at baseline, it was difficult to conduct meta-analysis with all the subgroups indicated above. Overall, six studies had high RoB and only three studies were evaluated as low overall RoB. Finally, even if we used extensively the search strategies to include all the potential studies, the researchers cannot exclude the possibility that some relevant studies have been missed out.

## Conclusion

Iron supplementation improved intelligence, attention and concentration, and memory of school-age children but did not affect school achievement. Iron supplementation benefitted the intelligence and memory of anemic school-age children, who are at higher risk of impaired cognitive development. Frequent iron administration had a greater effect on the intelligence and memory of the children, while short supplementation periods improved intelligence.

Our systematic review and meta-analysis showed overall that iron supplementation remains an effective intervention that can affect the cognition of school-age children. Further research should be encouraged in low-income settings using up-to-date, adapted and validated tools for the assessment of cognition.

## Supporting information

**S1 Checklist. PRISMA 2020 checklist.**
(DOCX)

**S1 Table. Medline search strategy for the effects of iron supplementation on cognitive development in school-age children.**
(DOCX)

**S2 Table. Embase search strategy for the effects of iron supplementation on cognitive development in school-age children.**
(DOCX)

**S3 Table. Scopus search strategy for the effects of iron supplementation on cognitive development in school-age children.**
(DOCX)

**S4 Table. Web of Science search strategy for the effects of iron supplementation on cognitive development in school-age children.**
(DOCX)

**S5 Table. Cochrane Library search strategy for the effects of iron supplementation on cognitive development in school-age children.**
(DOCX)

**S1 Fig. Frost plot, SMD analysis of the effect of iron supplementation on intelligence of school-age children (subgroup analysis for risk of bias of the study).**
(TIF)

**S2 Fig. Sensitivity analysis: Forest plot, SMD analysis of the effect of iron supplementation on intelligence of school-age children by excluding studies supplemented iron with folic acid or vitamin C.**
(TIF)

**S3 Fig. Frost plot, SMD analysis of the effect of iron supplementation on attention and concentration of school-age children (subgroup analysis for risk of bias of the study).**
(TIF)

**S4 Fig. Sensitivity analysis: Forest plot, SMD analysis of the effect of iron supplementation on attention and concentration of school-age children by excluding studies supplemented iron with folic acid or vitamin C.**
(TIF)

**S5 Fig. Frost plot, SMD analysis of the effect of iron supplementation on memory of school-age children (subgroup analysis for risk of bias of the study).**
(TIF)

**S6 Fig. Sensitivity analysis: Forest plot, SMD analysis of the effect of iron supplementation on memory of school-age children by excluding studies supplemented iron with folic acid or vitamin C.**
(TIF)

**S7 Fig. Frost plot, SMD analysis of the effect of iron supplementation on educational achievement of school-age children (subgroup analysis for risk of bias of the study).**
(TIF)

**S8 Fig. Forest plot, SMD analysis of the effect of iron supplementation on hemoglobin level of the school-age children (all included studies).**
(TIF)

**S9 Fig. Forest plot, SMD analysis of the effect of iron supplementation on hemoglobin level of the school-age children (categorized based on the cognitive domain).**
(TIF)

**S10 Fig. Forest plot, SMD analysis of the effect of iron supplementation on ferritin level of the school-age children.**
(TIF)

**S11 Fig. Forest plot, SMD analysis of the effect of iron supplementation on Serum transferrin level of the school-age children.**
(TIF)

## Acknowledgments

The authors are grateful to Dr. Ellen Deschepper from Ghent University Faculty of Medicine and Health Sciences for her support in the meta-analysis.

## Author Contributions

**Conceptualization:** Befikadu Tariku Gutema, Stefaan De Henauw, Souheila Abbeddou.

**Data curation:** Befikadu Tariku Gutema, Muluken Bekele Sorrie, Nega Degefa Megersa, Gesila Endashaw Yesera, Yordanos Gizachew Yeshitila, Souheila Abbeddou.

**Formal analysis:** Befikadu Tariku Gutema, Souheila Abbeddou.

**Investigation:** Befikadu Tariku Gutema, Souheila Abbeddou.

**Methodology:** Befikadu Tariku Gutema, Nele S. Pauwels, Souheila Abbeddou.

**Project administration:** Souheila Abbeddou.

**Software:** Befikadu Tariku Gutema, Nele S. Pauwels, Souheila Abbeddou.

**Supervision:** Nele S. Pauwels, Souheila Abbeddou.

**Validation:** Befikadu Tariku Gutema.

**Visualization:** Befikadu Tariku Gutema, Muluken Bekele Sorrie, Nega Degefa Megersa, Gesila Endashaw Yesera, Yordanos Gizachew Yeshitila.

**Writing – original draft:** Befikadu Tariku Gutema, Souheila Abbeddou.

**Writing – review & editing:** Befikadu Tariku Gutema, Muluken Bekele Sorrie, Nega Degefa Megersa, Gesila Endashaw Yesera, Yordanos Gizachew Yeshitila, Nele S. Pauwels, Stefaan De Henauw, Souheila Abbeddou.

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
