## [Decision Letter · Decision Letter 0]

14 Mar 2023

PONE-D-22-28873Effects of preventive iron supplementation on cognitive development in school-age children: Systematic Review and Meta-analysisPLOS ONE

Dear Dr. Gutema,

Thank you for submitting your manuscript to PLOS ONE. After careful consideration, we feel that it has merit but does not fully meet PLOS ONE’s publication criteria as it currently stands. Therefore, we invite you to submit a revised version of the manuscript that addresses the points raised during the review process.

We look forward to receiving your revised manuscript.

Kind regards,

Ammal Mokhtar Metwally, Ph.D (MD)

Academic Editor

PLOS ONE

“Funding: This work was conducted under the PhD studies of BTG, whose scholarship was partially funded by the Flemish Interuniversity Council (VLIR-UOS) in the context of the Institutional University Cooperation Program (IUC) with Arba Minch University https://www.vliruos.be/en/projects/project/22?pid=3604.”

4. "Thank you for stating the following financial disclosure:

“Funding: This work was conducted under the PhD studies of BTG, whose scholarship was partially funded by the Flemish Interuniversity Council (VLIR-UOS) in the context of the Institutional University Cooperation Program (IUC) with Arba Minch University https://www.vliruos.be/en/projects/project/22?pid=3604.”

5. Thank you for stating the following in the Acknowledgments/ Funding Section of your manuscript:

“This work was conducted under the PhD studies of BTG, whose scholarship was partially funded by the Flemish Interuniversity Council (VLIR-UOS) in the context of the Institutional University Cooperation Program (IUC) with Arba Minch University https://www.vliruos.be/en/projects/project/22?pid=3604.”

“Funding: This work was conducted under the PhD studies of BTG, whose scholarship was partially funded by the Flemish Interuniversity Council (VLIR-UOS) in the context of the Institutional University Cooperation Program (IUC) with Arba Minch University https://www.vliruos.be/en/projects/project/22?pid=3604.”

Additional Editor Comments:

The manuscript is interested meanwhile, the reviewers have raised a number of points which we believe would improve the manuscript and would allow a revised version to be published in PLOS one.

Reviewers' comments:

Reviewer's Responses to Questions

**Comments to the Author**

1. Is the manuscript technically sound, and do the data support the conclusions?

Reviewer #1: Yes

Reviewer #2: Yes

Reviewer #3: Yes

Reviewer #4: Yes

2. Has the statistical analysis been performed appropriately and rigorously? 

Reviewer #1: Yes

Reviewer #2: Yes

Reviewer #3: Yes

Reviewer #4: I Don't Know

3. Have the authors made all data underlying the findings in their manuscript fully available?

Reviewer #1: Yes

Reviewer #2: No

Reviewer #3: Yes

Reviewer #4: Yes

4. Is the manuscript presented in an intelligible fashion and written in standard English?

Reviewer #1: Yes

Reviewer #2: Yes

Reviewer #3: Yes

Reviewer #4: Yes

5. Review Comments to the Author

Reviewer #1: The topic of the study is very important as understanding the effects of iron supplementation on cognition among school age children is essential for the proper design of targeted strategies.

However, some aspects of the manuscript should be addressed:

Title:

“Effects of preventive iron supplementation on cognitive development in school-age children: Systematic Review and Meta-analysis”

I don’t understand what did the word (preventive) refer to? Did the authors intend to discuss iron supplementation to prevent iron deficiency anemia in school age children? I don’t think so, as the selected articles included anemic and non-anemic children.

Introduction

References in this section are not updated, most of them had been published one to two decades before.

This section lacks a paragraph illustrating the impact of iron deficiency status or iron deficiency anemia on cognitive development of school age children and consequently on their professional future.

Authors did not clarify the controversy between previous studies as regard the benefits and hazards of iron supplementation for this age group.

Objective

Is clear and well defined except for the word (preventive).

Methods

This section was written in a good technical standard and described in sufficient details. However, the part of statistical analysis was missed.

Methods of statistical analysis must be elaborated.

Readers must be prepared to understand the Forest plot figures which represented most of results. Values and significance of heterogeneity must be explained. If sensitivity analysis was applied or not.

Repeated sentences were detected in page 5 at lines 94-96.

As regards the definition of outcomes at page 5, I think no need for adding the biochemical outcomes as Hb concentration, serum ferritin level and transferrin saturation. These outcomes were not mentioned in the aims of the study and they will increase the diversity of results. Meanwhile, other outcomes as safety and adherence to the supplementation are closely related to the main outcomes.

Results

I was surprised that only one study out of the included 13 studies was performed in a low-income country, while anemia & iron deficiency are prevalent in these countries.

Comments on results were much confusing. These comments need rearrangement to rewrite the results of subgroups in a more structured manner. Authors can create a multilevel numbering to organize the comments on the main outcome domains and the underlying subgroups.

Results about safety outcome and adherence to the supplementation were missed.

Discussion

In the first page of discussion (page 20), the authors tried lengthily to prove that the studies included in the current review were also included in previous reviews, which added nothing to discussion.

The authors did not discuss explanation or hypothesis for the study findings. For example, the authors have to explain the underlying causes of non-improved school achievement in the studies included in this review, or non-improved cognitive functions in non-anemic participants, etc..

I think the authors need to revise the discussion to enrich it with causes of contradict or findings clarify the language, and adding updated references.

Strengths and weaknesses

These points were well documented.

Conclusions

Conclusions are presented in an organized and suitable manner.

Decision: Minor revision

Reviewer #2: We think this is a good systematic review and meta-analysis about the benefits of iron supplementation for the cognitive development in children. It has been well understood that in the school-aged children, iron supplementation has positive effects for improving age-adjusted height among anemic children, risk of anemia and iron deficiency, and the cognitive scores, including intelligence quotient, attention, and concentration.

Gutema et al has done a thorough work to apply their search strategy resulted in 6559 articles in the first phase and 1070 in the second phase. After which they screened again carefully and resulted in 13 articles included for the qualitative synthesis. We note that in 2022 there was a systematic review and meta-analysis published with the title of “Effect of Oral Iron Supplementation on Cognitive Function among Children and Adolescents in Low- and Middle-Income Countries: A Systematic Review and Meta-Analysis” which focused on studies in LMICs and in the age of children (and adolescents). Therefore, we acknowledge that Gutema et al implemented a different search strategy and inclusion criteria.

With the current title of “Effects of Preventive Iron Supplementation on Cognitive Development in School-Aged Children: Systematic Review and Meta-Analysis”, there are a few issues we would like to share:

1. The term ‘preventive’ used in the title does not seem to be suitable with the search strategy, analysis, and discussion. Page 26, Discussion section, line 389-390 also elaborated the aim of the review. Majority of the articles reviewed by Gutema et al recommended on the effects of iron in increasing the cognitive development parameters over 2 to 12 months supplementation. Therefore, we believe omitting the ‘preventive’ term would be more suitable.

2. The term ‘school-aged children’ does not match with the strategy applied by Gutema et al, because for example Gopaldas et al (1985 and 1987) included children aged 8-15 years, and Lynn and Harland (1998) included children aged 12-16 years.

We also would like to give input on some details below:

Page 8, line 39–40

Two keywords in the abstract: ‘iron supplementation’ and ‘school-aged children’ are not listed in the MeSH. We advise to choose the keywords based on the MeSH to increase the paper’s readability.

Page 10, line 81–82

“Studies conducted on children under five years or over 12 years of age were excluded.” However, Gopaldas et al (1985) and Kashyap and Gopaldas (1987) included children aged 8-15 years, Lynn and Harland (1998) included children aged 12-16 years, Sen and Kanani (2009) included children aged 9-13 years, Sungthong et al (2004) included children aged 6-13 years. Therefore, Gutema et al included studies in children aged more than 12 years old.

Page 10, line 82

The exclusion criteria applied by Gutema et al did not include other factors leading to bias, such as chronic morbidities and developmental disability. Whereas in Figure 1, there is an exclusion of 1 study due to Lead exposure. For this reason, we believe Gutema et al needs to add the exclusion criteria.

Page 10, line 87–89

“Observational study designs including case-report, case-series, ecological, cross-sectional, case-control, and cohort, quasi-experimental design, and reviews were excluded.”

This exclusion criteria is not needed because these other study designs are not included in the first place, as Gutema et al mentioned in the early paragraph that they only included RCTs.

Page 10 line 84-87

“Studies with multiple micronutrients are not included.” However, in page 14 line 178-180, some studies in combination with other supplementations (folic acid, albendazole, and ascorbic acid) were included in the review. While the studies involving EPA/DHA and multiple micronutrients were assessed properly by Gutema et al in separated groups.

Page 11 line 94–96

“The additional outcomes included anemia (hemoglobin concentrations below 115.0 g/l), hemoglobin concentrations (g/l), iron deficiency (measured using indicators of iron status including ferritin <15 μg/L and transferrin saturation <15%)”. We recommend the authors to mention the references for these cut off.

Page 11, line 106

“Screening in both phases was carried out by two reviewers (BTG, ND, MB, and GE), who independently included relevant articles and the above mentioned eligibility criteria. Disagreements were resolved by a third reviewer and consensus-based discussions. Screening of the updated records was carried out by one reviewer (BTG).”

We would like to confirm about the phases mentioned by the authors: Is the April search considered as the first phase, and the October search as the second phase? If they are, the authors need to inform about the reason for performing the second phase screening only by one person, while they stated that both phases involved two reviewers.

Page 19 line 220-221

“… daily 2 mg elemental iron”. This needs to be corrected to be 2 mg/kg, as is stated in the original paper, that is Soemantri (1989).

Table 2

Risk of bias for Sungthong (2004) indicated the paper has Low risk of bias in all domains (D1-D5). Figure 2 (Forest plot) suggested that iron daily and weekly were disadvantageous for cognitive development compared to placebo. We would suggest the authors to provide some possible explanations about this result, because it does not make sense for iron supplementations to have a more inferior outcome than ‘not giving iron supplementation’ (placebo). Gutema et al need to share their analysis about some factors in the study that could result in such unexpected outcome.

Table 2

The table indicated that Pollitt has a high overall risk of bias. Moreover, Figure 3 section D (iron deficient participants) displayed patients receiving iron supplementation has a lower SMD compared with the patients receiving placebo. We recommend Gutema et al to share their analysis about this result.

Figure 1

‘memroy’ is a typo, which is supposed to be memory.

Figure 3C

The figure showed that Pollit has only 100 patients with anemia and 1210 patients without anemia (in total is 1310 patients), whereas Table 1 mentioned that Pollit has 1775 subjects. Would it be possible there are significant numbers of missing data from Pollit?

Result section

Gutema et al occasionally mentioned the location name to refer to some particular studies, such as Kalibawang (when referring to Soemantri studies) and Vadodara (when referring to Sen and Kanani). I would suggest to mention the name of first author and year, i.e Soemantri (1985) than mentioning the place (Kalibawang).

Page 26, Discussion section

Gutema et al restated again about the inclusion and exclusion methods which we consider is unnecessary because these have been elaborated in the Results section.

Page 27 Line 419-420

Gutema et al need to mention ‘Figure 3’ as a reference for the description about prominent changes in the anemic groups.

Page 27 Discussion

For every aspect being compared, we believe the authors need to analyse about the reasons behind the results. The studies reviewed have different conclusions, which could be affected by the differences in population characteristics and dosages given. One example for this is the study by Sungthong (2004) which had significantly different outcomes compared with other studies, that is the placebo group had better outcomes than the iron-supplemented group.

Page 28 Strengths and Weaknesses of the Systematic Review and Meta-Analysis

From our perspectives, three limitations could be added, as follow:

1) Inability to conclude a specific dosage which would result in better cognitive development outcomes.

2) The observation period was relatively short: 2-3 months (in 5 studies), 4 months (in 5 studies), and 8-12 months (in 3 studies). The observation period in majority of studies could be too short for assessing a cognitive development.

3) The role of other micronutrients in some studies, that might affect the results.

Page 35, Figure 1

Records identified through database searching: 9041

Records after duplicates removed: 7669

Records screened: 6774

The authors need to provide information about what was done to reduce the number of articles from 7669 to 6774.

Reviewer #3: I have reviewed the study by Gutema et al.,

This systematic review and meta-analysis is an important study in its field. The high prevalence of iron deficiency related problems is still a burden in several developing countries. I have several things to be considered at:

1. There were several mistyping found in the text, I suggest the authors to double check the text again. One of the errors was memroy ("memory").

2. The systematic review process explained in the Discussion section (Line 392-407) should probably be written in Methods section as it was already written. My suggestion might need to be discussed further within the authors.

3. I did not find the Discussion section to be enticing as most of it was repeated sentences from Result section. I suggest the authors to consider putting more relevant things to be discussed or probably elaborate further with previous evidences.

Reviewer #4: I do not have any comments for the authors.

6. PLOS authors have the option to publish the peer review history of their article (what does this mean?). If published, this will include your full peer review and any attached files.

Reviewer #1: **Yes: **Ebtissam M. Salah El-Din

Reviewer #2: No

Reviewer #3: **Yes: **Surya Adhi

Reviewer #4: No

---

## [Author Response · Author response to Decision Letter 0]

27 Apr 2023

PONE-D-22-28873

Effects of iron supplementation on cognitive development in school-age children: Systematic review and Meta-analysis

Dear Editor

We thank the reviewers for their constructive comments, which helped us to improve our paper. Our responses to the reviewers are provided below, and all changes in the manuscript are marked with track changes. We hope that the revisions meet your expectations and that the manuscript will now be acceptable for publication.

Sincerely,

Befikadu Tariku Gutema

 

Reviewer #1:

Reviewer #1: The topic of the study is very important as understanding the effects of iron supplementation on cognition among school age children is essential for the proper design of targeted strategies.

However, some aspects of the manuscript should be addressed:

Title: “Effects of preventive iron supplementation on cognitive development in school-age children: Systematic Review and Meta-analysis”

I don’t understand what did the word (preventive) refer to? Did the authors intend to discuss iron supplementation to prevent iron deficiency anemia in school age children? I don’t think so, as the selected articles included anemic and non-anemic children.

Response: Thank you for the comments and suggestions. “Preventive” in this case means that iron supplementation was given only for the prevention not treatment. However, we agree with the reviewer that this could have also a different signification. The term ‘preventive’ is dropped now from the title. It reads now “Effects of iron supplementation on cognitive development in school-age children: Systematic Review and Meta-analysis”. 

Introduction

References in this section are not updated, most of them had been published one to two decades before.

Response: References were updated when possible. 

This section lacks a paragraph illustrating the impact of iron deficiency status or iron deficiency anemia on cognitive development of school age children and consequently on their professional future.

Response: We added three related statements on the impact of iron deficiency or iron deficiency anemia. Lines 65-66 read now “Iron deficiency and iron deficiency anemia can cause cognitive deficits (Jáuregui-Lobera, 2014; Samson, Fischer, & Roche, 2022).”

Authors did not clarify the controversy between previous studies as regard the benefits and hazards of iron supplementation for this age group.

Response: We added a statement regarding the safety of iron supplementation and the effect of iron supplementation on other parameters including hemoglobin concentration, anemia and iron deficiency. The text reads now in Lines 72-73 “Furthermore, iron supplementation was shown to be effective in improving cognition, safe in malaria settings, and had no gastrointestinal adverse effects.”

Objective

Is clear and well defined except for the word (preventive).

Response: The term preventive is dropped from the whole text.

Methods

This section was written in a good technical standard and described in sufficient details. However, the part of statistical analysis was missed.

Methods of statistical analysis must be elaborated.

Readers must be prepared to understand the Forest plot figures which represented most of results. Values and significance of heterogeneity must be explained. If sensitivity analysis was applied or not.

Response: We corrected by including additional information regarding the use Forest plot and sensitivity analysis. 

Repeated sentences were detected in page 5 at lines 94-96.

Response: Corrected, and the sentence is dropped.

As regards the definition of outcomes at page 5, I think no need for adding the biochemical outcomes as Hb concentration, serum ferritin level and transferrin saturation. These outcomes were not mentioned in the aims of the study and they will increase the diversity of results. Meanwhile, other outcomes as safety and adherence to the supplementation are closely related to the main outcomes.

Response: We planned to perform sub-group analysis based on the cut-off point for anemia, iron deficiency and iron deficiency anemia based on the availability of the data. That is why we defined the cut-off points based on the Hb concentration, serum ferritin and transferrin saturation. 

Results

I was surprised that only one study out of the included 13 studies was performed in a low-income country, while anemia & iron deficiency are prevalent in these countries.

Response: Indeed, we assume that this is because of the scarcity of studies assessing cognitive development in low-income countries. 

Comments on results were much confusing. These comments need rearrangement to rewrite the results of subgroups in a more structured manner. Authors can create a multilevel numbering to organize the comments on the main outcome domains and the underlying subgroups.

Response: Yes, there were long paragraphs which include the subgroup analysis part. We created addition layers of paragraphs based on the outcome with the number of subgroup analysis made for the domain. 

Results about safety outcome and adherence to the supplementation were missed.

Response: We added the results of safety and adherence to the supplementation. 

Discussion

In the first page of discussion (page 20), the authors tried lengthily to prove that the studies included in the current review were also included in previous reviews, which added nothing to discussion.

Response: we agree with the reviewer. This section has been dropped now.

The authors did not discuss explanation or hypothesis for the study findings. For example, the authors have to explain the underlying causes of non-improved school achievement in the studies included in this review, or non-improved cognitive functions in non-anemic participants, etc..

Response: We considered by adding statements reflecting the cause of non-improvement for some of the findings 

I think the authors need to revise the discussion to enrich it with causes of contradict or findings clarify the language, and adding updated references.

Response: Considered 

Strengths and weaknesses

These points were well documented.

Conclusions

Conclusions are presented in an organized and suitable manner.

Decision: Minor revision

Response: Thank you! 

 

Reviewer #2

We think this is a good systematic review and meta-analysis about the benefits of iron supplementation for the cognitive development in children. It has been well understood that in the school-aged children, iron supplementation has positive effects for improving age-adjusted height among anemic children, risk of anemia and iron deficiency, and the cognitive scores, including intelligence quotient, attention, and concentration.

Gutema et al has done a thorough work to apply their search strategy resulted in 6559 articles in the first phase and 1070 in the second phase. After which they screened again carefully and resulted in 13 articles included for the qualitative synthesis. We note that in 2022 there was a systematic review and meta-analysis published with the title of “Effect of Oral Iron Supplementation on Cognitive Function among Children and Adolescents in Low- and Middle-Income Countries: A Systematic Review and Meta-Analysis” which focused on studies in LMICs and in the age of children (and adolescents). Therefore, we acknowledge that Gutema et al implemented a different search strategy and inclusion criteria.

With the current title of “Effects of Preventive Iron Supplementation on Cognitive Development in School-Aged Children: Systematic Review and Meta-Analysis”, there are a few issues we would like to share:

1. The term ‘preventive’ used in the title does not seem to be suitable with the search strategy, analysis, and discussion. Page 26, Discussion section, line 389-390 also elaborated the aim of the review. Majority of the articles reviewed by Gutema et al recommended on the effects of iron in increasing the cognitive development parameters over 2 to 12 months supplementation. Therefore, we believe omitting the ‘preventive’ term would be more suitable.

Response: Thank you your thorough review and comment. The intention to use ‘preventive iron supplementation’ is to show the iron supplementation was given only for the prevention not treatment. Of course, if we consider supplementation (the primary intention is for apparently well population) is a dose for prevention. So, we dropped the word ‘preventive’ from the title, which reads now “Effects of iron supplementation on cognitive development in school-age children: Systematic Review and Meta-analysis”. 

2. The term ‘school-aged children’ does not match with the strategy applied by Gutema et al, because for example Gopaldas et al (1985 and 1987) included children aged 8-15 years, and Lynn and Harland (1998) included children aged 12-16 years.

Response: Yes, we considered school-age children. One of the exclusion criteria were if the article was based on only under five or only children above 12. These articles (Gopaldas et al (1985 and 1987), and Lynn and Harland (1998)) included children within our inclusion age categories but they also include beyond our age limit. To make it clear for the reader, we corrected the statement regarding the exclusion criteria by adding ‘only’ (page 4 line 87-88) to the age limits as follow: ‘…Studies conducted on children under five years only or over 12 years of age only,’. 

Page 8, line 39–40 

Two keywords in the abstract: ‘iron supplementation’ and ‘school-aged children’ are not listed in the MeSH. We advise to choose the keywords based on the MeSH to increase the paper’s readability.

Response: We changed ‘Iron supplementation’ to ‘Supplementations’ which is a Emtree for Embase. We could not find appropriate term for ‘school-aged children’. So, we opted to use as it is. 

Page 10, line 81–82

“Studies conducted on children under five years or over 12 years of age were excluded.” However, Gopaldas et al (1985) and Kashyap and Gopaldas (1987) included children aged 8-15 years, Lynn and Harland (1998) included children aged 12-16 years, Sen and Kanani (2009) included children aged 9-13 years, Sungthong et al (2004) included children aged 6-13 years. Therefore, Gutema et al included studies in children aged more than 12 years old.

Response: The response is indicated above. 

Page 10, line 82

The exclusion criteria applied by Gutema et al did not include other factors leading to bias, such as chronic morbidities and developmental disability. Whereas in Figure 1, there is an exclusion of 1 study due to Lead exposure. For this reason, we believe Gutema et al needs to add the exclusion criteria.

Response: It is documented that lead exposure is one of the causes for the decline in cognition and educational achievement. So, it showed that using the articles may affect the finding and conclusion because of the exposure rather than the supplementation. Based on your recommendation and to make it clear, we added an exclusion regarding to the exposure for the factors known to cause cognitive decline by adding the following phrase “….. or only on children exposed to known factors affecting cognitive decline (e.g. lead, arsenic) were excluded.”

Page 10, line 87–89

“Observational study designs including case-report, case-series, ecological, cross-sectional, case-control, and cohort, quasi-experimental design, and reviews were excluded.”

This exclusion criteria is not needed because these other study designs are not included in the first place, as Gutema et al mentioned in the early paragraph that they only included RCTs.

Response: We removed the statement. 

Page 10 line 84-87

“Studies with multiple micronutrients are not included.” However, in page 14 line 178-180, some studies in combination with other supplementations (folic acid, albendazole, and ascorbic acid) were included in the review. While the studies involving EPA/DHA and multiple micronutrients were assessed properly by Gutema et al in separated groups.

Response: Yes, we based on a sole supplementation of iron as an intervention. Regarding the EPA/DHA (27) and multiple micronutrients (37), at least one of the arms of the intervention was iron only. In addition, deworming was common practices for schoolchildren in place where helminth infection is prevalent, so we did not consider it as an additional intervention. Regarding iron folic acid is the common preparation form of iron supplementation and ascorbic acid was used as enhancer of the absorption of iron. We included this micronutrient and also did analysis based on the supplementation type separately as a subgroup. However, the issues of folic acid, ascorbic acid and deworming were not indicated in the criteria. So, we added the following statement to the ‘Eligibility criteria’ section. “However, iron with folic acid (one of the common supplement forms), iron with vitamin C (vitamin C was used as enhancer for absorption) and deworming were included in the study.”

Page 11 line 94–96

“The additional outcomes included anemia (hemoglobin concentrations below 115.0 g/l), hemoglobin concentrations (g/l), iron deficiency (measured using indicators of iron status including ferritin <15 μg/L and transferrin saturation <15%)”. We recommend the authors to mention the references for these cut off.

Response: We added references. 

Page 11, line 106

“Screening in both phases was carried out by two reviewers (BTG, ND, MB, and GE), who independently included relevant articles and the above mentioned eligibility criteria. Disagreements were resolved by a third reviewer and consensus-based discussions. Screening of the updated records was carried out by one reviewer (BTG).”

We would like to confirm about the phases mentioned by the authors: Is the April search considered as the first phase, and the October search as the second phase? If they are, the authors need to inform about the reason for performing the second phase screening only by one person, while they stated that both phases involved two reviewers.

Response: The first phase was the screening based on title and abstract, whereas the second phase was screening based on the full test based. Yes, we considered two reviewers to review for both phases. However, during the second-round search, which was carried out during the October 13th, 2022, we did the screening (for 1070 studies) using one reviewer. That was indeed an error and now corrected.

Page 19 line 220-221

“… daily 2 mg elemental iron”. This needs to be corrected to be 2 mg/kg, as is stated in the original paper, that is Soemantri (1989).

Response: Corrected 

Table 2

Risk of bias for Sungthong (2004) indicated the paper has Low risk of bias in all domains (D1-D5). Figure 2 (Forest plot) suggested that iron daily and weekly were disadvantageous for cognitive development compared to placebo. We would suggest the authors to provide some possible explanations about this result, because it does not make sense for iron supplementations to have a more inferior outcome than ‘not giving iron supplementation’ (placebo). Gutema et al need to share their analysis about some factors in the study that could result in such unexpected outcome.

Response: Yes, the result was controversial. We checked the magnitude of anemia (around 26%) and iron deficiency anemia (around 6%) among the study subject (on another publication). In this population, iron deficiency was not the major cause for anemia, so that the effect was also not insignificant or in another direction (may be over intake of iron due to supplementation where there was no deficiency). Now we included the characteristics of the study participants as a possible explanation for the finding. 

Table 2

The table indicated that Pollitt has a high overall risk of bias. Moreover, Figure 3 section D (iron deficient participants) displayed patients receiving iron supplementation has a lower SMD compared with the patients receiving placebo. We recommend Gutema et al to share their analysis about this result.

Response: The Pollitt study reported that the baseline value of IQ assigned to placebo were higher value than the iron supplementation group. Even if the study reported that there was no significant effect of iron supplementation, it also showed that the lower value was maintained for iron supplemented group compared to the placebo. On subgroup analysis, we used the endpoint value (unadjusted for baseline). Now we included a possible explanation for the finding.

Figure 1

‘memroy’ is a typo, which is supposed to be memory.

Response: Corrected 

Figure 3C

The figure showed that Pollit has only 100 patients with anemia and 1210 patients without anemia (in total is 1310 patients), whereas Table 1 mentioned that Pollit has 1775 subjects. Would it be possible there are significant numbers of missing data from Pollit?

Response: 1775 children were assessed and 417 were excluded. The total children included for the analysis were 1358 (101, 47 and 1210 children in the IDA, iron depleted, and iron-replete groups, respectively). The document did not indicate the exact number of children for each intervention group. So, we estimate half of the total sample for each group. We corrected the number in the table 1 to 1358. 

Result section

Gutema et al occasionally mentioned the location name to refer to some particular studies, such as Kalibawang (when referring to Soemantri studies) and Vadodara (when referring to Sen and Kanani). I would suggest to mention the name of first author and year, i.e Soemantri (1985) than mentioning the place (Kalibawang).

Response: Corrected 

Page 26, Discussion section

Gutema et al restated again about the inclusion and exclusion methods which we consider is unnecessary because these have been elaborated in the Results section.

Response: We removed the first two sentences as suggested.

Page 27 Line 419-420

Gutema et al need to mention ‘Figure 3’ as a reference for the description about prominent changes in the anemic groups.

Response: Corrected 

Page 27 Discussion

For every aspect being compared, we believe the authors need to analyse about the reasons behind the results. The studies reviewed have different conclusions, which could be affected by the differences in population characteristics and dosages given. One example for this is the study by Sungthong (2004) which had significantly different outcomes compared with other studies, that is the placebo group had better outcomes than the iron-supplemented group.

Response: Done 

Page 28 Strengths and Weaknesses of the Systematic Review and Meta-Analysis 

From our perspectives, three limitations could be added, as follow:

1) Inability to conclude a specific dosage which would result in better cognitive development outcomes.

Response: Done 

2) The observation period was relatively short: 2-3 months (in 5 studies), 4 months (in 5 studies), and 8-12 months (in 3 studies). The observation period in majority of studies could be too short for assessing a cognitive development.

Response: Done

3) The role of other micronutrients in some studies, that might affect the results.

Response: Done

Page 35, Figure 1

Records identified through database searching: 9041

Records after duplicates removed: 7669

Records screened: 6774

The authors need to provide information about what was done to reduce the number of articles from 7669 to 6774.

Response: We used DistillerSR for screening the articles. The gap between 7669 to 6774 was the number of articles uploaded during the 2nd round search and reviewed by one reviewer. Now we corrected it. 

 

Reviewer #3

Reviewer #3: I have reviewed the study by Gutema et al.,

This systematic review and meta-analysis is an important study in its field. The high prevalence of iron deficiency related problems is still a burden in several developing countries. I have several things to be considered at:

1. There were several mistyping found in the text, I suggest the authors to double check the text again. One of the errors was memroy ("memory").

Response: Corrected 

2. The systematic review process explained in the Discussion section (Line 392-407) should probably be written in Methods section as it was already written. My suggestion might need to be discussed further within the authors.

Response: Corrected 

3. I did not find the Discussion section to be enticing as most of it was repeated sentences from Result section. I suggest the authors to consider putting more relevant things to be discussed or probably elaborate further with previous evidences.

Response: Corrected 

 

Reviewer #4

Reviewer #4: I do not have any comments for the authors.

---

## [Decision Letter · Decision Letter 1]

30 May 2023

PONE-D-22-28873R1Effects of iron supplementation on cognitive development in school-age children: Systematic review and Meta-analysisPLOS ONE

Dear Dr. Gutema,

Thank you for submitting your manuscript to PLOS ONE. After careful consideration, we feel that it has merit but does not fully meet PLOS ONE’s publication criteria as it currently stands. Therefore, we invite you to submit a revised version of the manuscript that addresses the points raised during the review process.

We look forward to receiving your revised manuscript.

Kind regards,

Ammal Mokhtar Metwally, Ph.D (MD)

Academic Editor

PLOS ONE

Journal Requirements:

Reviewers' comments:

Reviewer's Responses to Questions

**Comments to the Author**

1. If the authors have adequately addressed your comments raised in a previous round of review and you feel that this manuscript is now acceptable for publication, you may indicate that here to bypass the “Comments to the Author” section, enter your conflict of interest statement in the “Confidential to Editor” section, and submit your "Accept" recommendation.

Reviewer #1: All comments have been addressed

Reviewer #2: All comments have been addressed

Reviewer #3: All comments have been addressed

2. Is the manuscript technically sound, and do the data support the conclusions?

Reviewer #1: Yes

Reviewer #2: Yes

Reviewer #3: Yes

3. Has the statistical analysis been performed appropriately and rigorously? 

Reviewer #1: Yes

Reviewer #2: Yes

Reviewer #3: Yes

4. Have the authors made all data underlying the findings in their manuscript fully available?

Reviewer #1: Yes

Reviewer #2: Yes

Reviewer #3: Yes

5. Is the manuscript presented in an intelligible fashion and written in standard English?

Reviewer #1: Yes

Reviewer #2: Yes

Reviewer #3: Yes

6. Review Comments to the Author

Reviewer #1: Thank you for giving me the opportunity to revise the re-submitted manuscript PONE-D-22-28873R1 under the title: “Effects of iron supplementation on cognitive development in school-age children: Systematic review and Meta-analysis”

Thanks for the authors for their thorough revision of the manuscript and their earnest response to most of reviewers’ comments. I enjoyed revising this manuscript, and believe that it is very promising.

• The authors corrected the title of the manuscript as most of reviewers have asked

• They have updated the section of introduction by adding new references (from 2014-2022)

• They have illustrated briefly the impact of iron deficiency on cognitive development.

• They have added a statement regarding the safety of iron supplementation, but they didn’t illustrate the possible hazards of iron supplementation for this age group. Daily iron supplementation can cause gastrointestinal symptoms as upset stomach, constipation, nausea, abdominal pain, vomiting, and diarrhea. High doses of iron might also cause more serious effects, including inflammation of the stomach lining and ulcers. Therefore, it is necessary to pay attention to the suitable formula and the appropriate effective dose to avoid serious side effects.

• The objective has been rephrased.

• I think the added information concerned with statistical analysis were not sufficient.

• Adding eligibility criteria has enriched the methodology section. However, inclusion of studies provided folic acid or vitamin C together with iron may affect the precision of analysis due to the known effect of folic acid on cognition and the enhancement of iron absorption induced by vitamin C.

• Though it was better to control the duration, frequency, and dose of iron supplementation to avoid the risk of bias. Practically, it is difficult to uniform the design of the available intervention trials.

• Authors have restructured results on the base of subgroup analysis, and have added the results of the previously missed parts about safety and adherence to the supplementation.

• They have modified the section of discussion to omit unnecessary paragraphs and add needed ones. However, they didn’t provide explanations for the unexpected outcomes. Some findings are still ambiguous (e.g., there was no effect of iron supplementation on the intelligence of children who were iron deficient at baseline. Iron-depleted children assigned to placebo, scored higher in RCPM test (IQ) at baseline compared to iron-depleted children who received the iron treatment [42]).

• I appreciate all authors efforts and consider this manuscript as an important addition to the field of child mental development.

Reviewer #2: I greatly appreciate the authors' effort to improve the manuscript. The authors have done a nice job of revising this manuscript and addressing the reviewers' questions/issues raised in the initial review. The revised manuscript is more logical; I am able to understand the methodology, study flow, and results. I recommend that the revised paper be accepted with very minor revisions. My comments below are related to the Conclusion presented in the paper.

I recommend the authors delete this sentence because it is irrelevant to the results, discussion, and study aim. Furthermore, the statement could be misunderstood as suggesting that a short duration of iron supplementation can be beneficial:

"Targeting anemic children, and a more frequent supplementation for shorter duration showed promising results."

Reviewer #3: I have reviewed the revised version of the manuscript from Gutema et al. I sincerely want to say thank you for the major improvements in the writings, especially in correcting for the typographical errors and other certain incorrect use of words in a sentence. The comments from other reviewer have also been addressed. However, I suggest to put the full form of abbreviated words as they first appeared on the text (e.g. RoB hasn't been explained until the end of the manuscript, the full word for RCPM hasn't been written) before this manuscript will be published.

The discussion sections is greatly improved and the authors have put great reasoning and comparations within their findings. I'm looking forward for this manuscript to be published as this study will provide important insights upon iron supplementation on cognitive development in children.

7. PLOS authors have the option to publish the peer review history of their article (what does this mean?). If published, this will include your full peer review and any attached files.

Reviewer #1: **Yes: **Ebtissam Mohamed Salah El-Din, Professor of Child Health, National Research Centre of Egypt

Reviewer #2: **Yes: **Cahyani Gita Ambarsari

Reviewer #3: **Yes: **Surya Adhi

---

## [Author Response · Author response to Decision Letter 1]

7 Jun 2023

Dear Dr. Ammal Mokhtar Metwally,

Thank you for your reply sent on May 31, 2023.

Enclosed we are submitting the revised manuscript PONE-D-22-28873R1, entitled “Effects of iron supplementation on cognitive development in school-age children: Systematic review and Meta-analysis”

By Befikadu Tariku Gutema, Muluken Bekele Sorrie, Nega Degefa Megersa, Gesila Endashaw Yesera, Yordanos Gizachew Yeshitila, Nele S. Pauwels, Stefaan De Henauw, and Souheila Abbeddou.

We thank the reviewers for their constructive comments, which helped us to improve our paper. Our responses addressing the concerns of the reviewers are provided below, and all changes in the manuscript are marked with track changes. We hope that the revisions meet your expectations and that the manuscript will now be acceptable for publication.

Sincerely,

Befikadu Tariku Gutema

 

Reviewer #1

Reviewer #1: Thank you for giving me the opportunity to revise the re-submitted manuscript PONE-D-22-28873R1 under the title: “Effects of iron supplementation on cognitive development in school-age children: Systematic review and Meta-analysis”

Thanks for the authors for their thorough revision of the manuscript and their earnest response to most of reviewers’ comments. I enjoyed revising this manuscript, and believe that it is very promising.

Thank you for your very positive feedback. We will address the remaining points. 

• The authors corrected the title of the manuscript as most of reviewers have asked

Indeed. 

• They have updated the section of introduction by adding new references (from 2014-2022)

Yes, we thank the reviewer as well from brining this up. 

• They have illustrated briefly the impact of iron deficiency on cognitive development.

Yes, that you. 

• They have added a statement regarding the safety of iron supplementation, but they didn’t illustrate the possible hazards of iron supplementation for this age group. Daily iron supplementation can cause gastrointestinal symptoms as upset stomach, constipation, nausea, abdominal pain, vomiting, and diarrhea. High doses of iron might also cause more serious effects, including inflammation of the stomach lining and ulcers. Therefore, it is necessary to pay attention to the suitable formula and the appropriate effective dose to avoid serious side effects.

Response: We agree that this part should be added to the discussion. Now added under the paragraph Lines 515-526 which reads: 

Concerns regarding possible side effects 

Iron supplementation can cause gastrointestinal side effects including loose stool/diarrhea, hard stool/constipation, abdominal pain, nausea, change in stool color, reflux/heartburn and headache [67]. Low doses of iron supplementation are reportedly not associated with an increased risk of infectious illness in children, except for a slight increase in the risk of developing diarrhea. Iron supplementation may increase the risk of malaria [reference 63], and the WHO recommends monitoring fever in preschool and school children receiving intermittent iron supplementation in malaria settings [22]. Despite the serious risk that is evidenced with iron supplementation, side effects were reported in two studies only [51, 52], while the increased risk of malaria infection was reported in one study only [48] and three studies indicated that the setting was malaria-free [42, 49, 51]. It is important that morbidity and potential side effects are assessed in studies and programs that supplement iron to children. 

• The objective has been rephrased.

Indeed, we rephrased it based on the previous revision.

• I think the added information concerned with statistical analysis were not sufficient.

Response: Now, on the tope of sensitivity analysis for the studies for trials with multiple form of supplementation of iron (frequency of administration or dose difference) we included to check iron supplementations combined with other micronutrients, and to check the effect of each study on the overall effect of the intervention by removing one trial at a time. 

• Adding eligibility criteria has enriched the methodology section. However, inclusion of studies provided folic acid or vitamin C together with iron may affect the precision of analysis due to the known effect of folic acid on cognition and the enhancement of iron absorption induced by vitamin C.

Response: These studies were included because of the form that is commonly distributed of iron with folic acid. But we agree that this could affect the analysis. We have carried out a sensitivity analysis and included the finding in the document. 

• Though it was better to control the duration, frequency, and dose of iron supplementation to avoid the risk of bias. Practically, it is difficult to uniform the design of the available intervention trials.

Response: Definitely, with the diverse design of the studies, it is almost impossible to uniform the findings. However, we tried in the subgroup analyses to provide at least an insight. 

• Authors have restructured results on the base of subgroup analysis, and have added the results of the previously missed parts about safety and adherence to the supplementation.

Thanks.

• They have modified the section of discussion to omit unnecessary paragraphs and add needed ones. However, they didn’t provide explanations for the unexpected outcomes. Some findings are still ambiguous (e.g., there was no effect of iron supplementation on the intelligence of children who were iron deficient at baseline. Iron-depleted children assigned to placebo, scored higher in RCPM test (IQ) at baseline compared to iron-depleted children who received the iron treatment [42]).

Response: The controversial results were driven by the study of Pollitt et al. (1989), where indeed baseline data were different between placebo and intervention groups. After conducting the same analysis without this study, the result showed significant improvement in cognition among iron deficient children in intelligence. We included the following in the result part. Lines 294-297 which reads: “The sensitivity analysis showed that iron supplementation significantly improved the intelligence of iron deficient children (SMD 0.52, 95%CI: 0.16, 0.87, P= 0.005, n=128; test for heterogeneity I2=0%, P=0.77) after dropping the study of Pollitt et al. (1989) [42].”

• I appreciate all authors efforts and consider this manuscript as an important addition to the field of child mental development.

We thank the reviewer and we hope that the additional revisions meet your expectations. 

 

Reviewer #2:

Reviewer #2: I greatly appreciate the authors' effort to improve the manuscript. The authors have done a nice job of revising this manuscript and addressing the reviewers' questions/issues raised in the initial review. The revised manuscript is more logical; I am able to understand the methodology, study flow, and results. I recommend that the revised paper be accepted with very minor revisions. My comments below are related to the Conclusion presented in the paper.

We thank the reviewer for their very positive feedback and for their comments that helped us improve the manuscript. 

I recommend the authors delete this sentence because it is irrelevant to the results, discussion, and study aim. Furthermore, the statement could be misunderstood as suggesting that a short duration of iron supplementation can be beneficial: "Targeting anemic children, and a more frequent supplementation for shorter duration showed promising results."

Response: The sentence has been dropped as recommended. 

 

Reviewer #3:

I have reviewed the revised version of the manuscript from Gutema et al. I sincerely want to say thank you for the major improvements in the writings, especially in correcting for the typographical errors and other certain incorrect use of words in a sentence. The comments from other reviewer have also been addressed. However, I suggest to put the full form of abbreviated words as they first appeared on the text (e.g. RoB hasn't been explained until the end of the manuscript, the full word for RCPM hasn't been written) before this manuscript will be published.

Response: Corrected. We added the abbreviation Under 'Methods and design in subtitle ‘Assessment of risk of bias’ part. The risk of bias (RoB)…. Regarding RCPM, the abbreviation was only at one point. The others were written in full. So, we replaced abbreviation by the full word. 

The discussion sections is greatly improved and the authors have put great reasoning and comparations within their findings. I'm looking forward for this manuscript to be published as this study will provide important insights upon iron supplementation on cognitive development in children.

We thank you for the very positive evaluation.

---

## [Editor Report · Decision Letter 2]

12 Jun 2023

Effects of iron supplementation on cognitive development in school-age children: Systematic review and Meta-analysis

PONE-D-22-28873R2

Dear Dr. Gutema,

We’re pleased to inform you that your manuscript has been judged scientifically suitable for publication and will be formally accepted for publication once it meets all outstanding technical requirements.

Kind regards,

Ammal Mokhtar Metwally, Ph.D (MD)

Academic Editor

PLOS ONE
---

## [Editor Report · Acceptance letter]

19 Jun 2023

PONE-D-22-28873R2 

Effects of iron supplementation on cognitive development in school-age children: systematic review and meta-analysis 

Dear Dr. Gutema:

I'm pleased to inform you that your manuscript has been deemed suitable for publication in PLOS ONE. Congratulations! Your manuscript is now with our production department. 

Kind regards, 

on behalf of

Professor Ammal Mokhtar Metwally 

Academic Editor

PLOS ONE